# SOD3 improves the tumor response to chemotherapy by stabilizing endothelial HIF-2α

Emilia Mira[1], Lorena Carmona-Rodríguez[1], Beatriz Pérez-Villamil[2], Josefina Casas[3], María Jesús Fernández-Aceñero[2], Diego Martínez-Rey[1], Paula Martín-González[1], Ignacio Heras-Murillo[1], Mateo Paz-Cabezas[2], Manuel Tardáguila[1,6], Tim D. Oury[4], Silvia Martín-Puig[5], Rosa Ana Lacalle[1], Gemma Fabriás[3], Eduardo Díaz-Rubio[2] & Santos Mañes[1]

One drawback of chemotherapy is poor drug delivery to tumor cells, due in part to hyperpermeability of the tumor vasculature. Extracellular superoxide dismutase (SOD3) is an antioxidant enzyme usually repressed in the tumor milieu. Here we show that specific SOD3 re-expression in tumor-associated endothelial cells (ECs) increases doxorubicin (Doxo) delivery into and chemotherapeutic effect on tumors. Enhanced SOD3 activity fostered perivascular nitric oxide accumulation and reduced vessel leakage by inducing vascular endothelial cadherin (VEC) transcription. SOD3 reduced HIF prolyl hydroxylase domain protein activity, which increased hypoxia-inducible factor-2α (HIF-2α) stability and enhanced its binding to a specific VEC promoter region. EC-specific HIF-2α ablation prevented both the SOD3-mediated increase in VEC transcription and the enhanced Doxo effect. SOD3, VEC, and HIF-2α levels correlated positively in primary colorectal cancers, which suggests a similar interconnection of these proteins in human malignancy.

[1] Department of Immunology and Oncology, Centro Nacional de Biotecnología/CSIC, Darwin, 3, Madrid, 28049, Spain. [2] Genomics and Microarray Laboratory, Medical Oncology & Surgical Pathology Departments, Instituto de Investigación Sanitaria San Carlos Hospital Clínico San Carlos, Univ. Complutense de Madrid, CIBERONC, Profesor Martín Lagos, S/N, Madrid, 28040, Spain. [3] Department of Biomedicinal Chemistry, Institute of Advanced Chemistry of Catalonia (IQAC-CSIC), Jordi Girona 18-26, Barcelona, 08034, Spain. [4] Department of Pathology, University of Pittsburgh, 3550 Terrace Street, Pittsburgh, PA 15261, USA. [5] Myocardial Pathophysiology Area, Centro Nacional de Investigaciones Cardiovasculares, Calle de Melchor Fernández Almagro, 3, Madrid, 28029, Spain. [6]Present address: Genetics Institute, University of Florida, 2033 Mowry Road, Gainesville, FL 32610, USA. E. Mira and L. Carmona-Rodríguez contributed equally to this work. Correspondence and requests for materials should be addressed to S.Mñe. (email: smanes@cnb.csic.es)

The endothelium forms the inner blood vessel barrier that regulates fluid, molecule and cell exchange between blood and interstitial tissues. From a molecular viewpoint, this endothelial barrier relies on several cell–cell adhesion systems in endothelial cells (ECs), including adherens junctions (AJ), and tight junctions (TJ), whose control is essential for vascular homeostasis[1]. In cancer, endothelial junction composition and function undergo extreme alterations that lead to massive edema and increased interstitial pressure; this reduces tumor perfusion and limits delivery of therapeutic agents[2,3]. Impaired blood flow in leaky vessels also aggravates hypoxia, which lends tumor cells greater resistance to anticancer compounds. Preservation of the endothelial barrier is thus of great clinical interest in oncology.

Vascular endothelial cadherin (VEC; cadherin 5, CD144) is responsible for endothelial AJ assembly and barrier architecture[1,4]. VEC mediates homophilic adhesion through cadherin repeats in its extracellular domain. The VEC intracellular region interacts with cytoplasmic proteins such as β-catenin and p120-catenin; these anchor VEC to the actin cytoskeleton and control VEC-elicited signals including modulation of the EC response to angiogenic factors, quiescence and polarity signals, and EC interaction with mural cells[1]. VEC also regulates transcription of the TJ protein claudin-5[5], which pinpoints VEC as a cornerstone in EC intercellular junction organization. In embryos and adults, slight changes in VEC function, localization, or expression severely destabilizes the vasculature[6]. Given its role in EC junction strength and plasticity, changes in VEC levels or trafficking are recurrently altered manifestations of the twisted, leaky blood vessel network in tumors[1,3].

Low oxygen tension, or hypoxia, is a hallmark of the tumor microenvironment and a major factor leading to EC dysfunction[7]. The adaptive cellular response to hypoxia is mediated by the basic helix-loop-helix/PERN-ARNT-SIM hypoxia-inducible transcription factors (HIF-1 and −2), heterodimers composed of HIF-α and -β subunits. HIF-β subunits are constitutively expressed and stable, whereas HIF-α subunits are regulated precisely by the HIF prolyl hydroxylase domain proteins (PHD1–3). In normoxic conditions, PHD enzymes hydroxylate two proline residues of HIF-α, which triggers binding of the Von Hippel-Lindau (VHL) E3 ubiquitin ligase to this subunit, and its subsequent ubiquitination and degradation[8,9]. In hypoxia, PHD become inactive, which stabilizes HIF-α subunits and triggers functional HIF heterodimers. Increased HIF levels in cancer cells induce overproduction of angiogenic factors such as vascular endothelial growth factor (VEGF), which in turn promotes angiogenesis and vessel leakiness[10]. Systemic postnatal ablation of *PHD2* induces hyperactive angiogenesis due to HIF-1 (but not HIF-2) stabilization in internal organs[11]. EC-specific partial reduction of PHD2 levels does not increase vascular density in tumors however, but tightens EC adhesion by increasing VEC transcription, which improves vascular function and chemotherapeutic drug delivery[12,13]. These effects are associated with HIF-2α (but not HIF-1α) stabilization in PHD2+/− haploinsufficient EC[12]; HIF-2α (also termed EPAS-1) but not HIF-1α induces VEC transcription[14].

Preclinical and clinical evidence shows that HIF-2α inhibitors reduce neoangiogenesis and growth of human renal clear cell carcinomas, a neoplasia characterized by VHL tumor-suppressor inactivation[15,16]. HIF-2α effects on the tumor milieu thus appear to be cell type specific and dose dependent, suggesting that cancer therapy can be enhanced by selective, precise HIF-2α stabilization in endothelial but not in cancer cells.

Nitric oxide (NO) is a regulator of PHD activity and hence of HIF-α levels. In normoxia, NO stabilizes HIF-1α in epithelial cells by inhibiting PHD[17]. Perivascular NO accumulation also reduces vascular permeability, increases tumor oxygenation, and improves response to radiotherapy[18]. NO levels must nonetheless be regulated precisely, since both inhibition and excess NO synthesis can induce AJ disassembly and vascular hyperpermeability[19,20]. These paradigmatic NO activities could be a result of its reaction with other environmental co-signals such as the superoxide free radical ($\cdot O_2^-$). $\cdot O_2^-$ concentrations not only determine NO steady-state levels but also the formation of highly reactive nitrogen species such as peroxynitrite ($\cdot ONOO^-$); indeed, HIF-α stability in human cerebral vascular smooth muscle cells (VSMCs) depends on NO and $\cdot O_2^-$ levels[21].

SOD3 (or extracellular superoxide dismutase) is a secreted enzyme that regulates the tissue redox balance by catalyzing $\cdot O_2^-$ dismutation to $H_2O_2$[22]. In contrast to intracellular SOD, SOD3 is expressed strongly in specific tissues and cells, including EC[23]. SOD3 has a tissue-protective role in the perivascular space by preventing oxidative damage of proteins and lipids and by preserving NO availability[24]. By controlling $NO/\cdot O_2^-$ reactions, SOD3 enhances NO-dependent vasorelaxation[25], which highlights its importance in vascular function. SOD3 effects in cancer biology, particularly on tumor vasculature, are poorly understood[26], although low SOD3 levels are associated with increased cancer incidence and poor prognosis[27–32]. Here we used pharmacological and genetic approaches to determine that restoration of SOD3 levels in tumors regulates tumor vasculature and increases the tumor response to chemotherapy.

## Results

**SOD3 upregulation potentiates doxorubicin effect on tumors**. We first analyzed whether SOD3 affects the tumor response to chemotherapy. SOD3 is usually downregulated in tumors compared to normal tissue[30]. Lovastatin (Lov) upregulates SOD3 in spontaneous mammary tumors in mice[33], which make Lov a useful strategy for inducing SOD3 in the tumor environment.

We generated subcutaneous tumors by injecting Lewis lung adenocarcinoma (LLC) cells in syngeneic wild-type (WT) and *SOD3*-deficient mice (SOD3−/−). SOD3−/− mice do not display any overt phenotype in basal conditions[22], and there were no major differences in EC marker expression compared to WT mice (Supplementary Fig. 1). Tumor-bearing mice were treated daily with a Lov dose equivalent to that in humans treated with 40 mg/day[34], or with vehicle (Vhcl) as control, starting at day 7 post-implant. A suboptimal doxorubicin (Doxo) dose was co-administered to mice. Although Lov alone did not affect tumor growth kinetics, it increased the Doxo antitumor effect in WT (Fig. 1a) but not in SOD3−/− mice (Fig. 1b). This enhancement correlated with a two-fold increase in Doxo levels in Lov-compared to Vhcl-treated WT mouse tumors (Fig. 1c); Lov treatment had no effect on Doxo levels in tumors in SOD3−/− mice.

Lov induced SOD3 in LLC tumors in WT but not in SOD3−/− mice (Supplementary Fig. 2a, b); this induction was also apparent in tumor sections (Fig. 1d). Although SOD3 is a soluble protein that can diffuse in the extracellular milieu, we observed a tendency to perivascular SOD3 accumulation in Lov-treated tumors grown in WT hosts. Staining of 3-nitrotyrosine (3-NT; a surrogate oxidative stress marker) in LLC tumors was reduced by Lov treatment in WT but not in SOD3−/− mice (Fig. 1e); these data linked the SOD3 increase to higher antioxidant activity in tumors from Lov-treated WT mice. To define the Lov-targeted cell types, SOD3 mRNA was quantified in LLC cells, leukocytes (CD45+), and ECs (CD31+) isolated from tumors grown in Vhcl- and Lov-treated WT and SOD3−/− mice. Lov minimally changed SOD3 mRNA in LLC cells in both mouse types (Fig. 1f) but induced a significant increase in leukocytes and ECs isolated from tumors in WT mice (Fig. 1f). This SOD3 upregulation was not observed in leukocytes or ECs from tumor-free spleen and lung

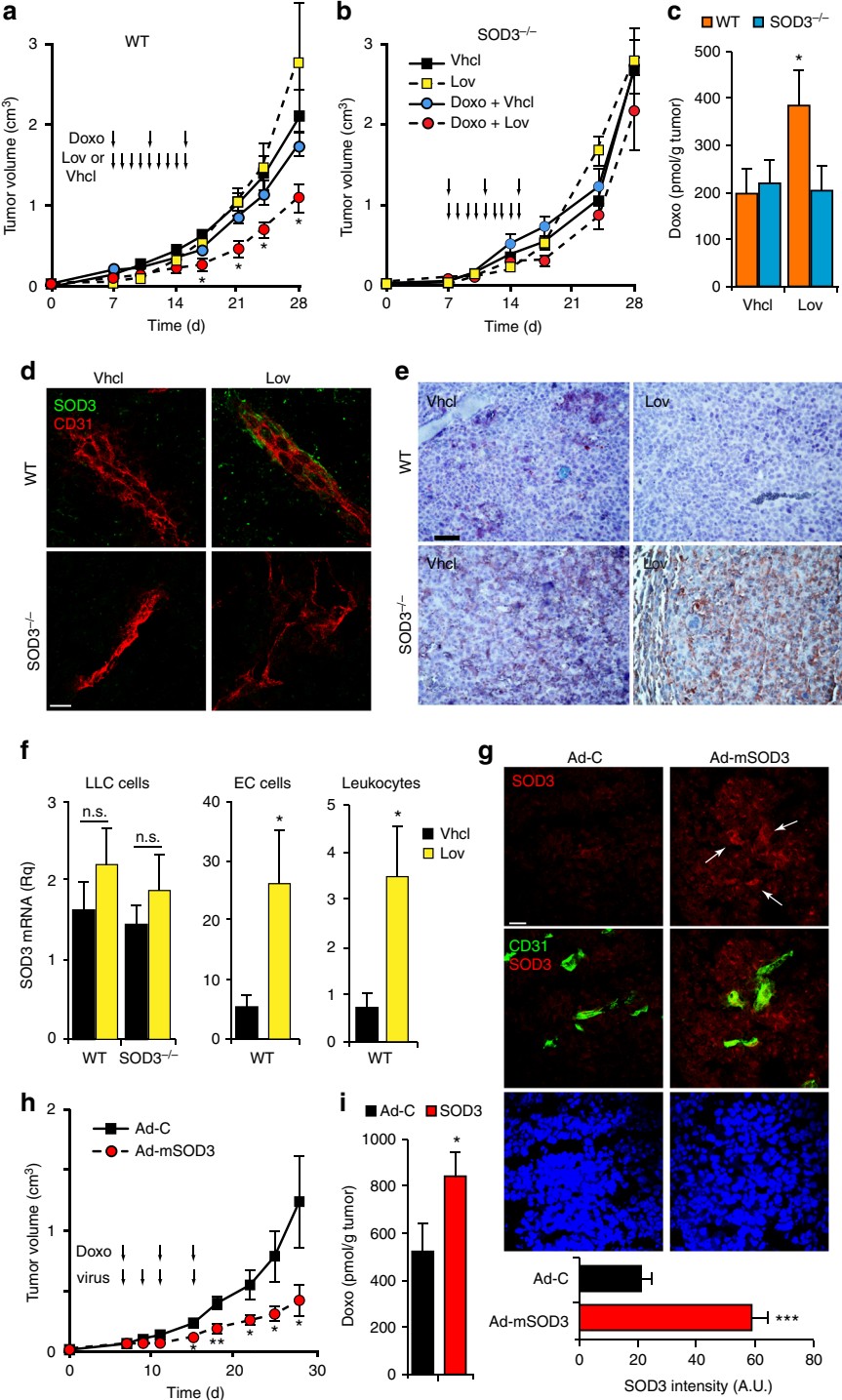

**Fig. 1** SOD3 upregulation enhances Doxo chemotherapeutic effects. **a**, **b** LLC tumor growth kinetics in Vhcl-, Lov-, Doxo+Vhcl-, or Doxo+Lov-treated WT (**a**) and SOD3$^{-/-}$ mice (**b**). Arrows indicate treatment schedule ($n = 10$ mice/group). **c** Tumors from WT or SOD3$^{-/-}$ mice treated as above were dissected on day 16 and Doxo was quantified in tumor extracts. **d** Vhcl- and Lov-treated tumors from WT or SOD3$^{-/-}$ mice were dissected on day 18 (<1 cm$^3$), and SOD3 and CD31 were detected in cryosections by immunohistochemistry (IHC); the two-color merge is shown ($n = 10$ fields/group; 4 mice/group). **e** 3-NT detection in paraffin sections of LLC tumors as in **d** ($n = 15$ fields/group; 4 mice/group). **f** LLC-GFP cells were implanted in WT or SOD3$^{-/-}$ mice, Lov-treated as in **a**, and tumors were dissected on day 21. LLC cells, ECs, and leukocytes were isolated by cell sorting and SOD3 mRNA was determined by qPCR. Data shown as mean ± SEM of triplicates ($n = 5$ mice/group). **g** Detection of SOD3 expression (red) and CD31 (green) in sections of Ad-C- or Ad-mSOD3-injected LLC tumors (dissected on day 18); nuclei are DAPI-stained (blue). Arrows in the SOD3 panel indicate the position of CD31$^+$ cells. Bottom panel shows SOD3 fluorescence intensity quantified by ImageJ (15 images/group; 4 mice/group). **h** Growth kinetics of Ad-C- or Ad-mSOD3-injected LLC tumors. Arrows indicate treatment schedule ($n = 9$ mice/group). **i** Doxo quantification in extracts of tumors dissected on day 16 from mice treated as in **h**. *$p < 0.05$, **$p < 0.01$, ***$p < 0.001$ one-way ANOVA with Dunnett's post-hoc test using Vhcl group as reference (**a**, **b**) or two-tailed Student's $t$-test (**f-i**). Bar, 10 μm (**d**, **g**) and 50 μm (**e**)

from the Lov-treated mice (Supplementary Fig. 3a) nor when Lov was added to the cultured 1G11 murine EC line or to 3T3 fibroblasts (Supplementary Fig. 3b, c). In our model, Lov induced SOD3 mainly in stromal cells in a tumor-specific manner.

Tumor necrosis factor (TNF)-α downregulates SOD3 in 3T3 cells[35] and Lov reduces TNF-α expression in tumor-infiltrating leukocytes[33]; Lov-induced SOD3 upregulation might be consequence of attenuation of inflammation in the tumor environment. In support of this idea, Lov reversed TNF-α-induced SOD3 inhibition in vitro in a dose-dependent manner (Supplementary Fig. 3d). TNF-α-induced SOD3 repression was also reversed by simvastatin but not by atorvastatin at the highest dose (Supplementary Fig. 3e). This suggests heterogeneity in the mechanism by which statins increase SOD3 levels.

**Forced SOD3 re-expression potentiates doxorubicin effect**. Statins are pleiotropic drugs with many effects on the vasculature and/or tumor cell growth[36]. To link SOD3 directly to Doxo effectiveness, we injected a recombinant adenovirus (Ad-mSOD3) to overexpress murine SOD3 in LLC tumors grafted into SOD3$^{-/-}$ mice; these experiments were done with no Lov co-treatment. Compared to the weak SOD3 staining in Ad-C (β-galactosidase-encoding)-injected tumors, Ad-mSOD3 injection notably increased SOD3 staining in tumor and stromal cells (Fig. 1g). Owing to the ability of Ad-mSOD3 to infect both tumor and stromal cells, SOD3 production was distributed homogeneously throughout the tumor parenchyma. In the absence of Doxo, Ad-mSOD3- and Ad-C-injected tumor growth was comparable (Supplementary Fig. 4a), which suggested that SOD3 overexpression had no inherent protumor or antitumor activity. Ad-mSOD3 injection nonetheless enhanced the Doxo inhibitory effect on tumor growth (Fig. 1h), which again correlated with higher Doxo levels in Ad-mSOD3- compared to Ad-C-injected tumors (Fig. 1i). Pharmacological or genetic SOD3 upregulation in the tumor environment thus increased levels of the small compound Doxo in tumors.

**Intratumor SOD3 levels regulate EC structure and function**. We tested whether SOD3 upregulation altered perfusion of the tumor parenchyma. Blood vessel labeling with fluorescein isothiocyanate (FITC)-conjugated lectin showed that Ad-mSOD3 treatment increased the percentage of lectin$^+$CD31$^+$ vessels compared to Ad-C (Fig. 2a, Supplementary 4b), which implicates SOD3 directly in promoting tumor perfusion. Despite increased perfusion in Ad-mSOD3-injected tumors, we noted an unanticipated reduction in blood vessel number (Fig. 2b). Mean vessel area was nonetheless increased (Fig. 2c), which implied greater tumor irrigation. Analysis of CD31 staining in thick sections from Ad-C- and Ad-mSOD3-injected tumors (Fig. 2d) showed that SOD3 overexpression increased vessel length and diameter (Fig. 2e, f), although vessel branching was unaffected (Fig. 2g).

We also observed an increase in the percentage of vessels double stained with lectin and CD31 in Lov- compared to Vhcl-treated WT mice (Fig. 2h). When Lov and Doxo treatments were combined, perfusion also increased in WT mice (Supplementary Fig. 4c). In SOD3$^{-/-}$ mice, Lov treatment did not significantly enhance tumor perfusion compared to Vhcl, although there was a clear trend toward a higher percentage of lectin$^+$CD31$^+$ vessels in these mice (Fig. 2h). Lov treatment might thus have additional, SOD3-independent effects on tumor perfusion, possibly linked to its pleiotropic activities on the endothelium. Analysis of lectin$^+$ vessels in thick sections from Vhcl- and Lov-treated tumors (Fig. 2i) indicated that, as observed in Ad-mSOD3-injected tumors, Lov treatment of WT mice increased mean vessel area and tumor vessel length (Fig. 2j, k), although blood vessel

diameter decreased compared to Vhcl-treated mice (Fig. 2l). These differences were not observed in tumors from Lov- and Vhcl-treated SOD3$^{-/-}$ mice, which indicates SOD3 influence on these changes. Vessel branching was comparable in all mice.

Scanning electron microscopy (SEM) suggested that Lov treatment altered vessel ultrastructure in tumors grown in WT mice, as lumens were smoother than those in Vhcl-treated WT or Lov-treated SOD3$^{-/-}$ mice (Fig. 2m). Lov treatment reduced tumor vessel leakage (dextran extravasation) in WT compared to Vhcl-treated but not in SOD3$^{-/-}$ mice (Fig. 2n, o). Alteration in VEC expression is sufficient to disrupt EC intercellular junctions[1]. Lov-treated tumors in WT mice had higher VEC mRNA and protein levels than tumors from Vhcl-treated WT or Lov-treated SOD3$^{-/-}$ mice (Fig. 2p–r). The extension of VEC junctions increased in tumor vessels from Lov- compared to Vhcl-treated WT mice (4.7 ± 0.5 μm vs. 3.1 ± 0.2 μm; $p < 0.05$, two-tailed Student's $t$-test; $n = 55$). This was not observed in SOD3$^{-/-}$ mice (Vhcl, 3.8 ± 0.3 μm, $n = 83$; Lov, 3.1 ± 0.2 μm, $n = 38$).

VEC-stained area in CD31$^+$ structures was larger in Ad-mSOD3- than in Ad-C-injected tumors (Supplementary Fig. 4d,e). Ad-mSOD3 also enlarged the extension of VEC junctions compared to Ad-C (Ad-mSOD3, 7.1 ± 1.6 μm, $n = 50$; Ad-C, 2.9 ± 0.5 μm, $n = 81$; $p < 0.05$, two-tailed Student's $t$-test), which indicates that SOD3 directly regulates VEC expression in tumor-associated ECs.

**SOD3 upregulation in ECs enhances intratumor Doxo levels**. We analyzed whether specific endothelial SOD3 expression enhanced tumor response to chemotherapy. To generate a conditional mouse strain, we crossed a loxP-SOD3KI mouse line (Fig. 3a) with an EC-specific inducible Cre-driver line (VE-Cad-Cre$^{ERT2}$)[37]. Isogenic littermates, Cre$^+$ (SOD3 overexpression in EC; SOD3$^{EC-Tg}$) or Cre$^-$ (no ectopic SOD3 expression), were inoculated with LLC cells, and EC-SOD3 overexpression was induced by two tamoxifen injections; higher tamoxifen doses impaired tumor angiogenesis, independently of SOD3 induction. Tamoxifen induced strong SOD3 expression in ~50% of tumor vessel ECs in Cre$^+$ but not in Cre$^-$ mice (Fig. 3b). The Doxo antitumor effect was greater in SOD3$^{EC-Tg}$ than in Cre$^-$ littermates (Fig. 3c), and intratumor Doxo levels were higher in SOD3$^{EC-Tg}$ mice (Fig. 3d). The percentage of lectin-perfused blood vessels was higher in SOD3$^{EC-Tg}$ tumors compared to controls (Fig. 3e, f), which suggested improved tumor perfusion after EC-SOD3 expression. Analysis of CD31 indicated that EC-SOD3 expression reduced the total number of CD31-stained vessels (Fig. 3g, h), although the vessel-covered area was larger in SOD3$^{EC-Tg}$ tumors (Fig. 3i). SOD3$^{EC-Tg}$ tumor vessels were larger and of greater diameter than those in control mice (Fig. 3j, k), whereas vessel branching was unaffected by SOD3 (Fig. 3l).

The VEC-stained area relative to that of CD31 (Fig. 3m, n) and the extension of VEC junctions also increased in SOD3$^{EC-Tg}$ vessels compared to controls (SOD3$^{EC-Tg}$, 2.7 ± 0.4 μm, $n = 83$; Cre$^-$, 1.5 ± 0.1 μm, $n = 130$; $p < 0.05$, two-tailed Student's $t$-test). The reduced dextran-red leakage in SOD3$^{EC-Tg}$ vessels (Fig. 3o) also suggests a tighter EC barrier. Specific EC-SOD3 expression is sufficient to improve the LLC tumor response to chemotherapy and to increase EC barrier firmness.

**SOD3 induces VEC expression via a NO-dependent pathway**. We studied the mechanism by which SOD3 stimulates *VEC* gene transcription. The microvascular 1G11 endothelial cell line[38] was transduced with a bicistronic retrovirus expressing green fluorescent protein (GFP) and SOD3. Enhanced SOD3 expression was verified by immunoblot of lysates from stable mock and

SOD3 transfectants (Supplementary Fig. 5a), although most SOD3 was secreted to the medium (Supplementary Fig. 5b). Dihydroethidium oxidation, used to measure $\cdot O_2^-$ levels, was lower in 1G11-SOD3- than in 1G11-mock-transduced cells, both in basal culture and after incubation with tumor cell-conditioned medium, a stress condition that boosted $\cdot O_2^-$ production in

mock-transfected cells (Supplementary Fig. 5c). 1G11-SOD3 cells thus overexpress enzymatically functional SOD3.

SOD3 overexpression in 1G11 cells upregulated VEC expression compared to mock cells, as detected by immunoblot and reverse transcriptase quantitative PCR (RT-qPCR) (Fig. 4a, b); SOD3-induced VEC expression was also evident by

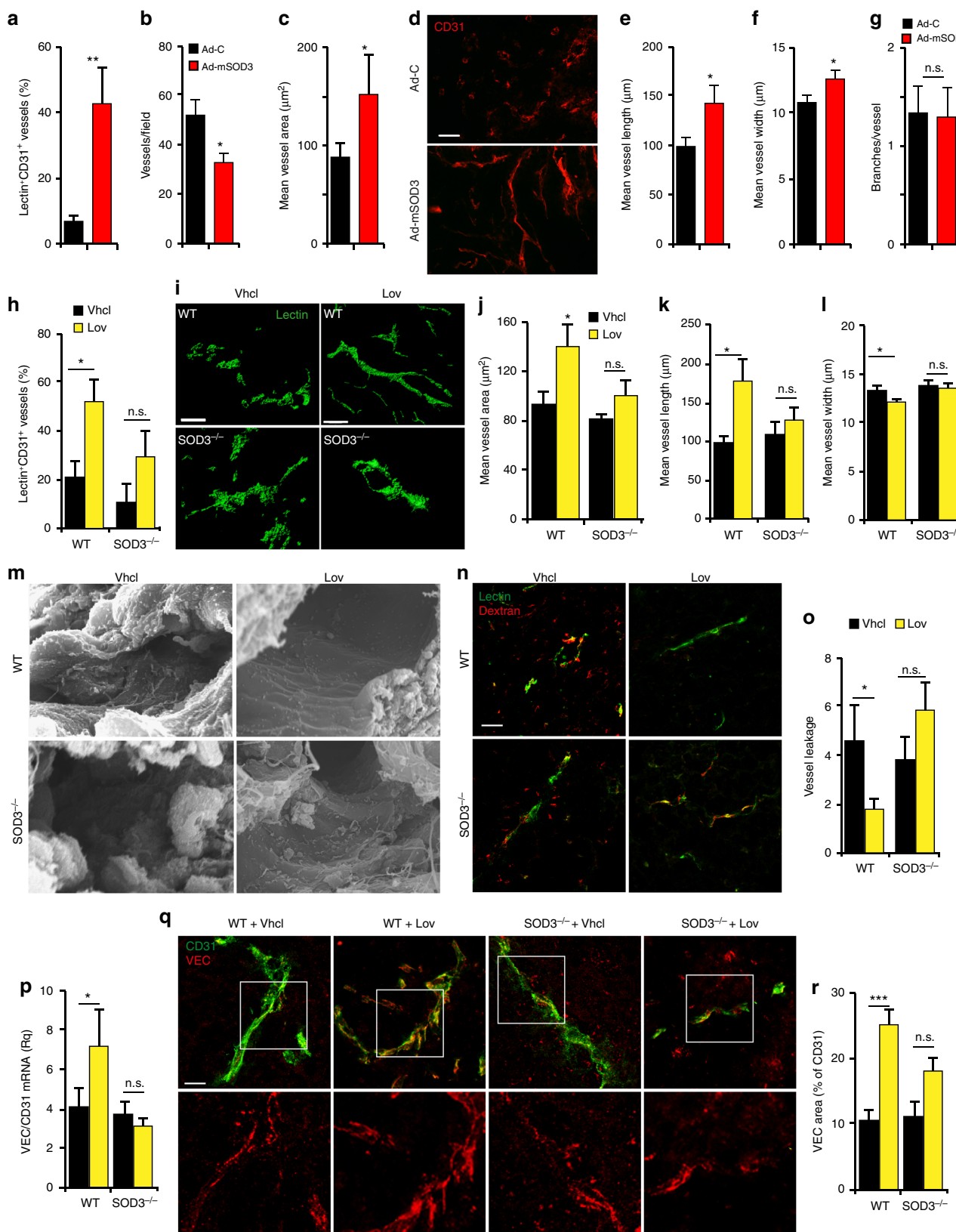

immunofluorescence (IF) (Fig. 4c). Quantification of continuous and discontinuous AJ[39] indicated that SOD3 increased the percentage of continuous VEC junctions (Fig. 4d) and reduced the number of intercellular gaps (Fig. 4e); these data suggest that SOD3 promotes EC interconnectivity. Permeability to dextran-FITC particles was lower in 1G11-SOD3 monolayers (Fig. 4f), which supports a role for SOD3 in vascular integrity. Over-expression of the inactive SOD3[S195C] mutant[40] did not reduce 1G11 permeability (Fig. 4f), although mutant and WT enzyme expression were comparable (Fig. 4g). These results indicate the need for SOD3 enzyme activity to regulate EC permeability. SOD3 silencing in 1G11 cells reduced VEC expression, as detected by IF (Supplementary Fig. 6a–d). This reduction in VEC levels was associated with increased dextran-FITC permeability of SOD3-silenced 1G11 cell monolayers compared to controls (Supplementary Fig. 6e).

To further test whether SOD3 increased barrier stability, we analyzed permeability to dextran-FITC particles in VEGF-treated 1G11-SOD3 and mock cells. In accordance with its leakiness-inducing activity, VEGF increased permeability in 1G11-mock monolayers; this did not occur in 1G11-SOD3 cells (Fig. 4h). VEC staining (Fig. 4i) indicated that VEGF-treated mock cells showed a reduction in the length of continuous VEC junctions (Fig. 4j) and an increase in intercellular gaps (Fig. 4k); VEGF minimally affected these parameters in 1G11-SOD3 cells. The SOD3-induced increase in VEC expression thus counteracted VEGF-induced vessel leakiness.

SOD3 scavenging activity increases NO availability[24]; fluorescence-activated cell sorting (FACS) analysis of the NO probe DAF2 showed higher intracellular NO levels in 1G11-SOD3 than in -mock cells (Fig. 4l, m). Analysis of NO distribution using the NO-sensitive probe 4,5-diamino-rhodamine B (DAR-1) in LLC tumors indicated that SOD3 induction by Lov increased DAR-1-associated fluorescence (formed after the NO reaction) near vascular structures in tumors from WT but not from SOD3[−/−] mice (Fig. 4n). A total of 76% of FITC-lectin[+] structures in WT tumors ($n = 46$) showed more intense DAR-1 staining than in adjacent parenchyma, whereas strong DAR-1 staining was seen in only 50% of SOD3[−/−] tumor vessels ($n = 34$; $p = 0.016$, Pearson's $\chi^2$). These differences in the DAR-1 staining pattern cannot be attributed to a differential host-dependent Lov effect on NO synthesis, since endothelial NO synthetase (eNOS) expression in tumor-associated vasculature was comparable in Lov-treated WT and SOD3[−/−] mice (Fig. 4o), also observed in WT and SOD3[−/−] mice in basal conditions (Supplementary Fig. 1). SOD3 levels might influence perivascular NO levels in vitro and in vivo.

To further analyze the role of NO in the SOD3-mediated increase in EC barrier function, we treated 1G11-mock and -SOD3 cells with the NOS inhibitor L-NMMA prior to permeability analysis. Whereas Vhcl-treated 1G11-SOD3 mono-layers were less permeable than those of 1G11-mock cells, L-NMMA treatment increased 1G11-SOD3 permeability to levels similar to controls (Fig. 4p). Moreover, whereas SOD3 over-expression in 1G11 cells increased luciferase activity driven by a ~2.5 kb VEC promoter, L-NMMA treatment reduced SOD3-induced VEC promoter activity to control levels (Fig. 4q). To confirm NO as a direct regulator of VEC transcription, we incubated VEC-luciferase reporter-transfected 1G11 cells with the NO donor (Z)-1-[N-(2-aminoethyl)-N-(2-ammonioethyl) amino] diazen-1-ium-1,2-diolate (DETA-NONOate). Low DETA-NONOate doses increased VEC promoter-driven lucifer-ase activity compared to Vhcl-treated cells, whereas very high DETA-NONOate concentrations inhibited VEC-driven tran-scription (Fig. 4r). In accordance with this biphasic NO effect on VEC promoter activity, low and high DETA-NONOate concentrations had opposite effects on 1G11 permeability to dextran-FITC (Fig. 4s). A dose-dependent NO-mediated mechan-ism thus appears to underlie SOD3-induced VEC transcription and EC barrier stability in 1G11 cells.

SOD3 overexpression in human dermal microvasculature-derived EC (HDMEC) and in bovine aortic EC (BAEC) also increased VEC mRNA and protein levels and reduced perme-ability to dextran-FITC in a NO-dependent manner (Supple-mentary Fig. 7). SOD3 effects on VEC expression and the EC barrier are therefore not restricted to mouse-derived ECs or due to clonal artifacts.

**HIF-2α mediates SOD3 enhancement of VEC transcription.** To identify the transcription factor involved in SOD3 induction of VEC transcription, we analyzed a series of truncated VEC pro-moters in mock- and SOD3-transfected 1G11 cells. SOD3 potentiated transcription driven by ~2.5 and ~1 kb VEC pro-moters but not of those <0.5 kb (Fig. 5a), which indicated critical SOD3 regulatory elements at the −500/−1000 region. This region has three hypoxia response elements (HRE) and one KLF-4-binding site (Fig. 5b). HRE are HIF-1 and HIF-2 consensus binding sites. Only HIF-2α overexpression induced VEC promoter-driven transcription of a reporter gene (Supplementary Fig. 8a, b). These results support the reported specific HIF-2α role in VEC transcription[14]. Chromatin immunoprecipitation (ChIP) with anti-HIF-2α and -KLF-4 antibodies showed SOD3-induced association of HIF-2α (but not KLF-4) to the VEC promoter −500/−1000 region (Fig. 5c, ChIP2); HIF-2α or KLF-4 interaction at more distal promoter sites was nonetheless SOD3 independent (Fig. 5c, ChIP1).

Using directed mutagenesis, we determined the role of each HRE in the region critical for SOD3-induced VEC transcription. Elimination of the HRE1 site did not alter SOD3-mediated

**Fig. 2** SOD3 upregulation alters tumor-associated vasculature. **a** Lectin-FITC-perfused, CD31-stained vessels in Ad-C and Ad-mSOD3-injected tumors (tumor size 0.5–1.2 cm³). Data are shown as the percentage of CD31[+]lectin[+] vessels relative to total CD31[+]-stained vessels. See also Supplementary Fig. 4b for representative images. **b**, **c** Density (**b**) and mean area (**c**) of CD31[+] structures in Ad-C- and Ad-mSOD3-injected tumors at end point. **d** CD31 staining (red) of sections from Ad-C- and Ad-mSOD3-injected LLC tumors. **e–g** Determination of the mean length (**e**), diameter (**f**), and branches per vessel (**g**) from images as in **d** ($n \geq 60$ vessels/condition). **h** Percentage of CD31[+]lectin[+] vessels relative to total CD31[+]-stained vessels in Vhcl- and Lov-treated LLC tumors (1–2.5 cm³) grafted into WT and SOD3[−/−] mice. **i** Reconstruction of tumor blood vessels from Vhcl- or Lov-treated, lectin-perfused WT and SOD3[−/−] mice using the Imaris software (tumor sections 50 μm). **j** Mean vessel area in tumors from Vhcl- and Lov-treated WT and SOD3[−/−] mice determined from images stained with anti-CD31 antibody. **k**, **l** Determination of mean length (**k**) and diameter (**l**) of CD31-stained tumor vessels from Vhcl- or Lov-treated WT and SOD3[−/−] mice ($n \geq 70$ vessels/condition). **m** Scanning electron microscopic analysis of blood vessels in tumors from Vhcl- and Lov-treated WT and SOD3[−/−] mice dissected at day 20 (tumor size <1 cm³). **n** Vessel leakiness in LLC tumors dissected at day 16 (size <1 cm³), determined by FITC-lectin and Texas Red-dextran injection, with two-color merge (yellow). **o** Vessel permeability of samples as in **n**, determined as the ratio of fluorescence intensity of extravasated dextran to perfused vessel counts. **p** Determination of VEC mRNA levels, normalized to those of CD31 (EC marker), in tumors from Vhcl- and Lov-treated WT and SOD3[−/−] mice. **q** Images of VEC (red) and CD31 (green) staining in tumors from Lov-treated WT or SOD3[−/−] mice dissected at day 20. Bottom panels show only VEC staining in a magnified area. **r** VEC-stained area, determined as a percentage of VEC staining in CD31[+] structures, in images as in **q**. In all cases, 10–20 images were analyzed from ≥5 mice/group; *$p< 0.05$, **$p< 0.01$, ***$p< 0.001$; two-tailed Student's $t$-test. Bar, 50 μm

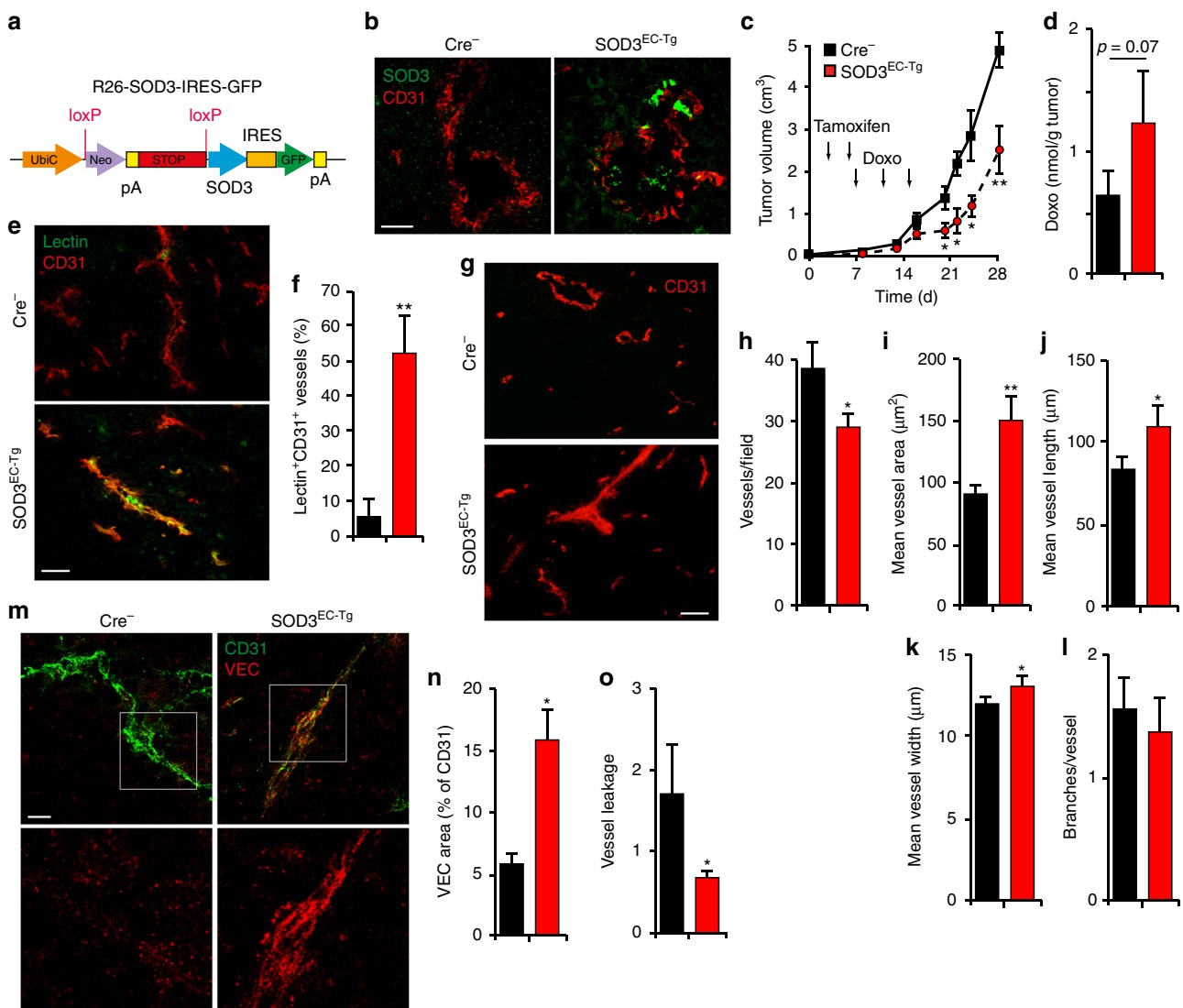

**Fig. 3** EC-specific SOD3 expression is sufficient to increase Doxo effect. **a** Conditional expression allele for the SOD3-IRES-GFP gene at the ROSA26 locus. **b** SOD3 (green) and CD31 (red) detection in sections from tumors grown in Cre-expressing and Cre⁻ mice after tamoxifen induction. **c** LLC tumor growth kinetics in SOD3^EC-Tg and control (Cre⁻) mice. Arrows indicate tamoxifen and Doxo treatments ($n = 10$/group). **d** Doxo quantification in extracts of tumors dissected at day 16 from control and SOD3^EC-Tg mice ($n = 15$ or 17 mice/group). **e, f** Images and quantification of lectin-FITC-perfused and CD31-stained vessels in control and SOD3^EC-Tg tumors (0.7–1.2 cm³) dissected at day 20. **g** CD31 staining of control and SOD3^EC-Tg tumors at end point. **h–l** CD31⁺ structure density (**h**), mean area covered by CD31⁺ structures (**i**), mean vessel length (**j**) and diameter (**k**), and the number of vessel branches (**l**), determined in images as in **g**. **m** CD31 (green) and VEC (red) staining of tumors grafted in SOD3^EC-Tg and control mice. Bottom panels show red staining in a magnified area. **n** Quantification of the VEC area in CD31⁺ structures from images as in **m**. **o** Vessel permeability determined by FITC-lectin and Texas Red-dextran in tumors dissected at day 17 from control and SOD3^EC-Tg mice. For **b**, **e–o**, 10–20 images were analyzed from 5 mice/group; $*p < 0.05$, $**p < 0.01$, two-tailed Student's $t$-test. Bar, 50 µm

enhancement of VEC promoter-driven activity, whereas mutation of HRE2 or HRE3 sites was sufficient to abolish the SOD3 transcription effect (Fig. 5d). These results identify SOD3 as a positive regulator of HIF-2α-mediated VEC transcription, probably by fostering HIF-2α association to HRE2 and HRE3 sites on the VEC promoter.

To further link HIF-2α to SOD3-induced VEC transcription, we used short hairpin RNA (shRNA) to stably silence this factor in 1G11 cells (Fig. 5e). HIF-2α knockdown reduced VEC mRNA levels in 1G11 cells (Fig. 5f); VEC levels in HIF-1α-silenced cells were unchanged (Supplementary Fig. 8c, d), which supports a specific HIF-2α role in VEC transcription. SOD3 also activated the VEC promoter (1 kb) in shNT-1G11 (control) but not in shHIF-2α-silenced cells (Fig. 5g), which indicates that SOD3-induced VEC promoter activity is HIF-2α dependent. SOD3

increased luciferase activity driven by a synthetic promoter bearing 9× HRE sites (Supplementary Fig. 9a), which pinpoints SOD3 as a general activator of HIF-2α-mediated transcription.

**SOD3 stabilizes HIF-2α by curbing PHD activity.** We examined how SOD3 regulates HIF-2α activity. SOD3 overexpression did not appreciably change HIF-2α mRNA (Supplementary Fig. 9b) but increased HIF-2α protein levels (Fig. 6a). SOD3 not only augmented total HIF-2α staining but also increased the area and intensity of HIF-2α-stained nuclei in 1G11 (Fig. 6b–d) and in HDMEC (Supplementary Fig. 9c–e). Gradual HIF-2α overexpression triggered proportional VEC transcription (Supplementary Fig. 9f). SOD3-induced HIF-2α stabilization might thus explain SOD3-induced VEC transcription.

As NO negatively regulates PHD activity[17], we tested whether SOD3 regulates HIF-2α posttranslationally. Levels of PHD-2 (the most abundant isoform) were comparable in 1G11-SOD3 and -mock cells (Supplementary Fig. 9g), which suggested that SOD3 does not affect PHD abundance. The selective PHD-2 inhibitor IOX2[41] increased VEC promoter-driven luciferase activity in 1G11-mock vs. Vhcl-treated cells but did not induce higher VEC promoter activity in 1G11-SOD3 cells than that observed after

SOD3 overexpression (Fig. 6e); this implies that IOX2 and SOD3 act on the same pathway to boost VEC promoter-driven transcription.

To quantify PHD activity in mock- and SOD3-transfected cells, we established an enzyme-linked immunosorbent assay (ELISA) to detect proline hydroxylation of a peptide from the HIF-1α oxygen-dependent degradation domain, with recombinant HA-tagged VHL (Fig. 6f). VHL discriminated between hydroxylated

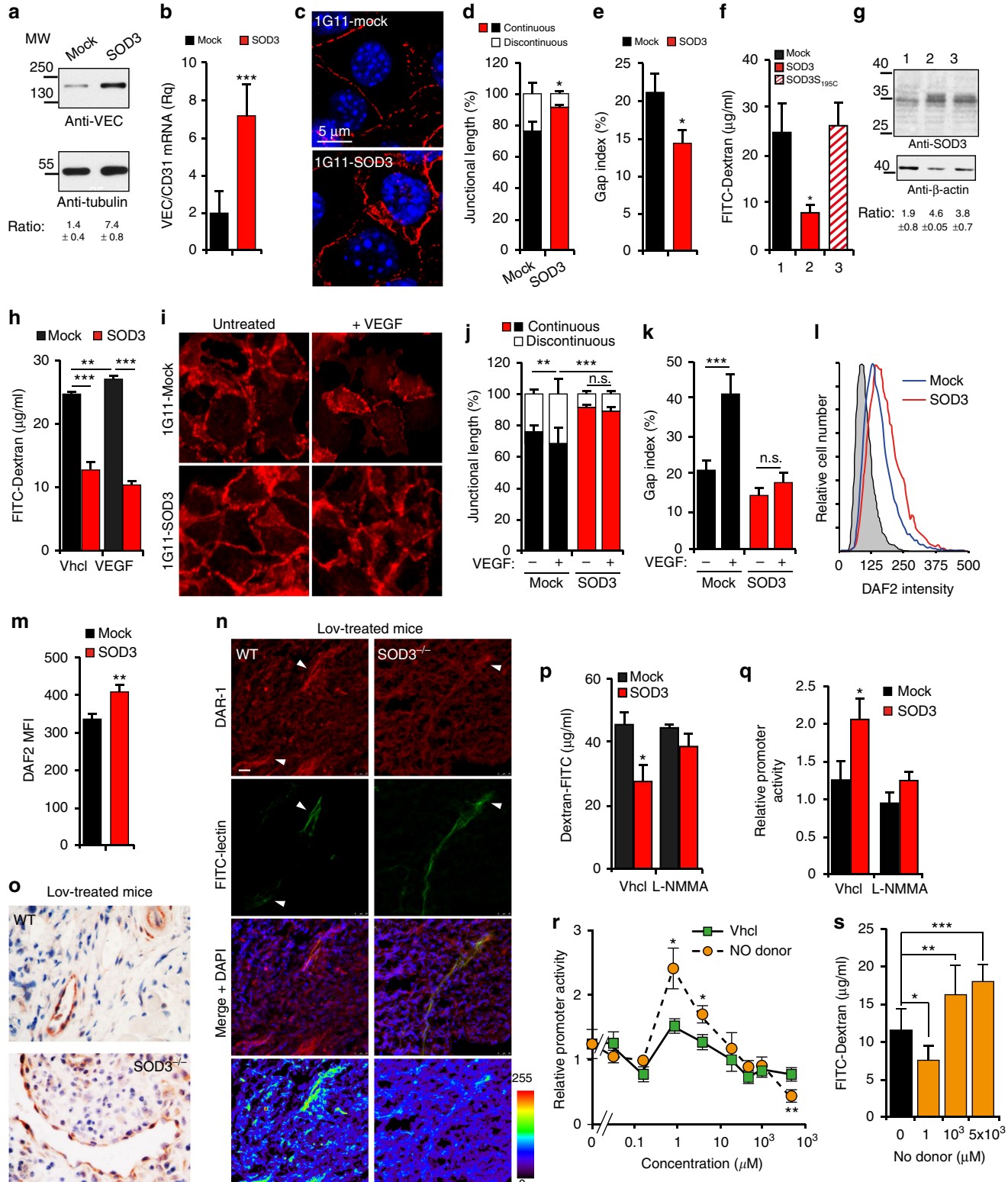

and non-hydroxylated HIF-1α peptides (Supplementary Fig. 9h), which confirmed ELISA specificity. SOD3 overexpression reduced peptide hydroxylation (Fig. 6g), which was associated with reduced HIF-2α ubiquitination in 1G11-SOD3 cultured in normoxia (Fig. 6h). HIF-2α degradation was delayed after re-oxygenation of hypoxic cultured 1G11-SOD3 compared to mock cells (Fig. 6i, j); oxygen-induced HIF-1α degradation was unaffected by SOD3 overexpression (Supplementary Fig. 9i, j). Our data showed more rapid HIF-1α decay after re-oxygenation than HIF-2α, which agrees with differential HIF-1α and HIF-2α sensitivity to oxygen-induced breakdown in ECs[12]. The degree of SOD3-induced PHD inhibition might be insufficient to prevent this rapid HIF-1α degradation.

The nuclear HIF-2α-stained area and intensity were reduced in L-NMMA- than in Vhcl-treated 1G11-SOD3 cells (Fig. 6k–m), which implicates NO in SOD3-mediated HIF-2α stabilization. DETA-NONOate treatment increased VEC promoter-driven luciferase activity in HIF-2α- but not in mock-transfected 3T3 fibroblasts (Supplementary Fig. 9k). As 3T3 cells in normoxia do not express HIF-2α, our data indicate that NO induces VEC promoter-driven transcription by regulating HIF-2α levels.

**HIF-2α ablation in EC impedes SOD3 effects on Doxo activity.** To confirm the role of endothelial HIF-2α in mediating SOD3 effects on tumor vasculature, we analyzed the result of SOD3 overexpression on Doxo effectiveness in mice lacking HIF-2α in ECs (HIF-2α$^{EC-KO}$), obtained by crossing HIF-2α$^{f/f}$[42] with VE-Cad-Cre$^{ERT2}$ mice. HIF-2α and VEC mRNA levels were reduced in EC isolated from tumors grafted in HIF-2α$^{EC-KO}$ mice vs. Cre$^-$ littermates after tamoxifen injection (Supplementary Fig. 10a, b), which supports HIF-2α as a VEC regulator in tumor endothelium.

To determine whether the improved, SOD3-induced response to chemotherapy is HIF-2α dependent, we injected Ad-C or Ad-mSOD3 virus and Doxo into tumor-bearing tamoxifen-treated HIF-2α$^{EC-KO}$ or Cre$^-$ mice. Coinciding with our earlier data (Fig. 1h), Ad-mSOD3 increased Doxo activity in controls compared to Ad-C treatment (Fig. 7a), while the Ad-mSOD3 effect was lost when LLC tumors were grafted into HIF-2α$^{EC-KO}$ mice (Fig. 7b). SOD3 overexpression improved perfusion of tumors implanted in control but not in HIF-2α$^{EC-KO}$ mice (Fig. 7c, d). Ad-mSOD3-injected tumors in controls but not in HIF-2α$^{EC-KO}$ mice reduced blood vessel number (Fig. 7e) but increased the tumor area covered by vessels as well as their length and diameter (Fig. 7f–h); the branching pattern was unaffected (Fig. 7i). The SOD3-mediated increase in VEC mRNA and protein levels in controls was not observed in HIF-2α$^{EC-KO}$ mouse tumors (Fig. 7j, k; Supplementary Fig 10c–f). The data indicate that HIF-2α is central to SOD3-induced VEC

transcription and enhanced Doxo antitumor activity. Our results suggest that SOD3 improves tumor vascular function by augmenting HIF-2α-dependent VEC expression, probably by increasing NO availability and thus inhibition of PHD activity.

**SOD3 correlates with VEC and HIF-2α levels in human cancers.** We next measured SOD3 and VEC mRNA levels in freshly frozen samples from a cohort of 102 colorectal carcinoma (CRC) patients (stages I–IV) and in non-tumor samples >10 cm from the primary tumor in the same patients. We found lower SOD3 and VEC mRNA levels in CRC than in non-tumor samples (Fig. 8a, b), with a positive correlation of levels for these genes in CRC samples (Fig. 8c). SOD3 mRNA reduction was not associated to a specific stage or CRC molecular subtype[43].

To further study this SOD3/VEC association, we analyzed an independent validation cohort comprised of 87 CRC stage III and 19 non-tumor samples. SOD3 and VEC mRNA levels were again lower in CRC than in non-tumor samples (Fig. 8d, e), with positive correlation between these genes in tumor samples (Fig. 8f). The results thus suggest SOD3 and VEC coregulation in CRC.

Since SOD3 regulates HIF-2α posttranscriptionally in mouse ECs, we tested the association between SOD3 and HIF-2α by immunohistochemistry (IHC) in the epithelial tumor cell compartment and tumor-associated stroma, using a tissue microarray (TMA) of 89 CRC samples (stages I–IV; Fig. 8g); 23.6% of tumor and 8.2% of stromal samples showed no SOD3 or HIF-2α staining. We estimated separate $H$-scores for each marker in each tumor area; values correlated positively for SOD3 and HIF-2α in stroma (Fig. 8h) and negatively in the tumor compartment (Fig. 8i). SOD3 and HIF-2α might thus be regulated distinctly in epithelial and stromal cells.

As SOD3 regulated HIF-2α stability in mouse EC, we studied the expression of these two genes in the CRC TMA, using CD34 as a label of tumor-associated ECs; 55.4% of CD34$^+$ ECs were SOD3 stained (Fig. 8j; Supplementary Fig. 11). Some sections showed adjacent SOD3-stained and unstained ECs, which implies that this pattern was not artifactual. Most SOD3-stained ECs also showed nuclear HIF-2α accumulation (97.6%). These data suggest that SOD3 stabilizes HIF-2α in human tumor-associated ECs, similar to observations in our mouse tumor models.

## Discussion

The success of chemotherapy relies on the delivery of sufficient amounts of cytotoxic drugs to kill tumor cells. Progressing tumors have aberrant, hyperpermeable vasculature, which reduces tumor perfusion and restricts diffusion of small molecules from the bloodstream to the tumor interstitial space[2,3]. Here we identify

**Fig. 4** SOD3-induced VEC expression requires SOD3 enzyme activity and NO. **a** VEC and tubulin (loading control) levels in 1G11-mock- and -SOD3-transduced cells. The VEC/tubulin ratio is indicated ($n = 3$). **b** VEC mRNA normalized to CD31 levels in mock and SOD3-expressing cells ($n = 5$). **c** VEC staining (red) in 1G11-mock- and 1G11-SOD3-transduced cells. Nuclear staining with DAPI (blue) ($n > 20$ fields/condition). **d** Continuous vs. discontinuous junctions expressed as junction length (% total junction length) in mock and SOD3-expressing cells. **e** Gap index in confluent cells as in **c**. (**d**, **e**, $n \geq 5$ fields/condition). **f** FITC-dextran (40 kDa) permeability of 1G11-mock, -SOD3 and -SOD3$^{S195C}$ cell monolayers ($n = 3$). **g** SOD3 levels in cells as in **f**; bottom, filter rehybridized with actin. SOD3/actin ratio indicated ($n = 3$). **h** FITC-dextran permeability of 1G11-mock and -SOD3 monolayers, untreated or VEGF-pretreated ($n = 9$). **i** VEC staining of untreated and VEGF-treated cells as in **h**. **j**, **k** Continuous vs. discontinuous junctions (**j**) and gap index (**k**) in VEGF-treated and untreated cells; $n \geq 5$ fields/condition. **l** FACS analysis of DAF2 staining in 1G11-mock and -SOD3 cells. Non-specific staining (gray). **m** DAF2 intensity from cells as in **l** ($n = 3$). **n** NO distribution in LLC tumors from Lov-treated WT and SOD3$^{-/-}$ mice, visualized by DAR-1, microangiography using FITC-lectin, merge of DAR-1, FITC-lectin, and DAPI counterstaining, and pseudocolor representation of DAR-1 microfluorographs. Arrowheads indicate large blood vessels. Images are confocal microscopy z-stacks. **o** eNOS detection in sections from tumors as in **n**. **p** FITC-dextran permeability of L-NMMA- or Vhcl-pretreated 1G11-mock and -SOD3 monolayers ($n = 6$). **q** VEC promoter activity in L-NMMA- or Vhcl-treated 1G11 cells ($n = 5$). **r** VEC promoter activity in DETA-NONOate- or Vhcl-treated 1G11 cells ($n = 9$). **s** FITC-dextran permeability of DETA-NONOate-pretreated 1G11-mock and -SOD3 monolayers; 0 = Vhcl ($n = 9$). Data shown as mean ± SEM. *$p < 0.05$, **$p < 0.01$, ***$p < 0.001$; two-tailed Student's $t$-test (**b**, **d**, **e**, **k**, **m**, **p–r**) or ANOVA with Dunnett's (**f**, **s**) or with Tukey's (**h**, **j**) post-hoc tests. Bar, 5 μm (**c**) and 25 μm (**i**, **n**)

SOD3 upregulation as a target for improving tumor perfusion and selective Doxo delivery through increased VEC transcription via HIF-2α. Elevated SOD3 levels, but not those of a catalytically inactive mutant, reduced monolayer permeability and prevented the VEGF-induced destabilization of AJ. Specific HIF-2α ablation in EC prevented these SOD3 activities, which suggests HIF-2α as a critical SOD3 mediator in vivo.

Reports are contradictory regarding SOD3 influence on tumor progression and angiogenesis. SOD3 not only mediates VEGF-C-induced angiogenesis and metastasis in a specific breast cancer subset[44] but also inhibits angiogenesis by reducing VEGF-A expression[45] or release from the extracellular matrix[46]. A positive feedback loop between Ras activation and SOD3 is suggested to boost cancer cell proliferation[26,47], although SOD3 is down-regulated via transcriptional[29,30] and posttranscriptional mechanisms[48] in a number of human cancers. We found SOD3 mRNA downregulation in two independent CRC cohorts compared to healthy colon tissue, which coincides with low SOD3 protein levels in CRC biopsies[32]. SOD3 silencing thus appears to be common in several malignant tumors.

We used three approaches to upregulate SOD3 in the in vivo tumor environment, one pharmacological (Lov administration) and two genetic (intratumor Ad-mSOD3 injection and EC-specific inducible SOD3 expression). Statins are pleiotropic drugs that can affect the endothelium through various mechanisms[36]. Given that Lov did not improve Doxo delivery or induce VEC expression in tumors implanted in SOD3−/− mice, these Lov effects could be considered SOD3 dependent. Lov upregulated SOD3 significantly in ECs and leukocytes isolated from LLC tumors implanted in WT hosts but not in ECs and leukocytes from tumor-free organs from these mice. These data indicate that Lov upregulates SOD3 in stromal cells via an indirect mechanism that operates in the tumor microenvironment. Our results show that Lov and simvastatin reversed the TNF-α-induced inhibition of SOD3 expression; these statins might thus induce SOD3 by attenuating inflammation. Atorvastatin, added alone or with TNF-α, did not increase SOD3 mRNA. Individual statins can induce specific gene expression profiles, which has been associated with their intracellular concentration[49]. Whether the disparate effects on SOD3 expression between these statins are due to bioavailability or mechanistic differences requires further study.

Another pending question is whether SOD3-induced vascular remodeling depends on its expression in a specific tumor compartment. Lov induced SOD3 in ECs and leukocytes. SOD3 overexpression in ECs was sufficient to increase Doxo effectiveness and reduce tumor vessel permeability, although we observed similar effects after intratumor injection of Ad-mSOD3, which infects and triggers SOD3 expression in tumor and stromal cells. SOD3 is a soluble, secreted enzyme and hence can diffuse in the tumor tissue. Elevation of SOD3 to a level that markedly reduces oxidative radicals in the tumor milieu might thus normalize tumor vasculature as efficiently as perivascular SOD3 expression.

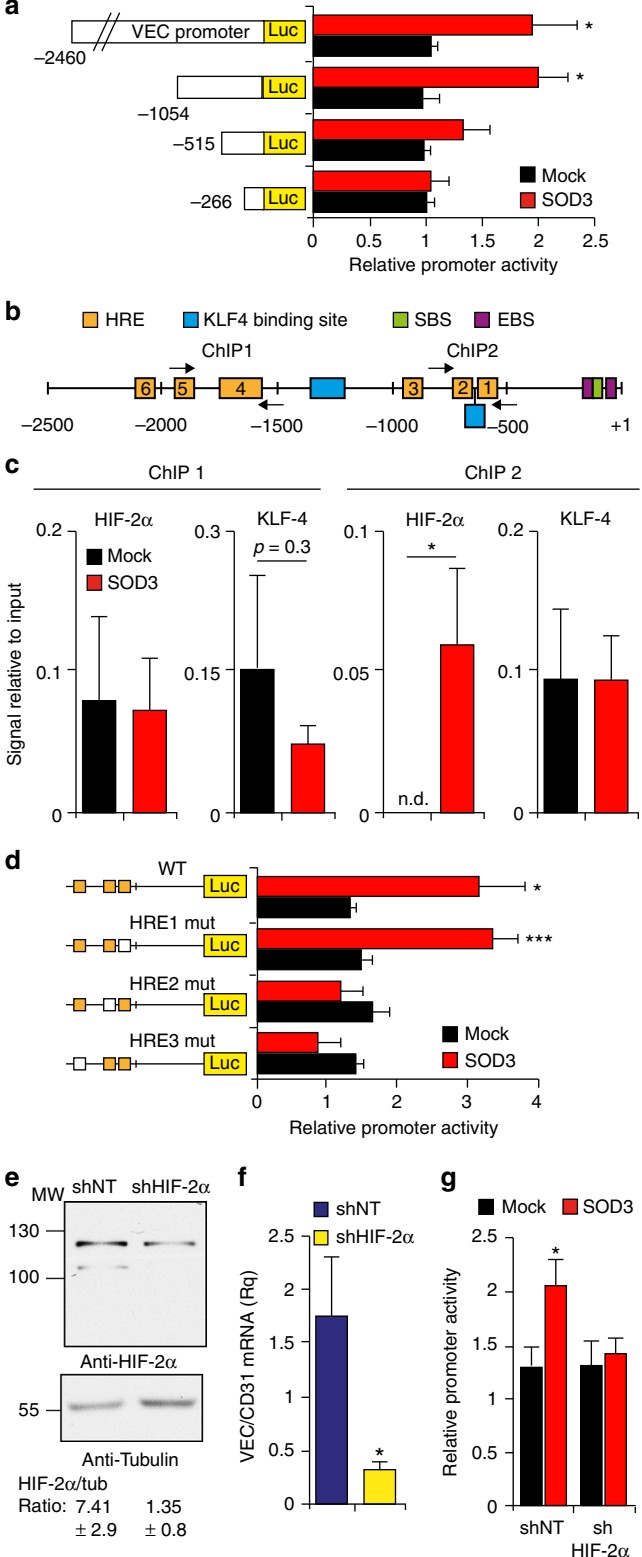

**Fig. 5** HIF-2α mediates SOD3-induced VEC transcription. **a** Relative luciferase activity (RLU) in 1G11 cells cotransfected with SOD3 (or mock) and the full-length VEC promoter or deletion mutants. Values were normalized to the signal in 1G11-mock cells for each promoter (n = 4). **b** Full-length (~2.5 kb) VEC promoter with binding sites for the main transcription factors. Numbers indicate positions relative to the transcription start site. HRE hypoxia response element, SBS Sox-binding site, EBS Ets-1-binding site. **c** ChIP analysis of VEC promoter in mock- and SOD3-transfected 1G11 cells using anti-HIF-2α and -KLF-4 antibodies. Coprecipitated DNA fragments were PCR-quantified using primers in **a** (arrows; n = 5). **d** RLU in 1G11-mock or -SOD3 cells transfected with VEC promoter HRE mutants. The mutated HRE site is indicated (white square; n = 10). **e** Immunoblot analysis of control (shNT) and HIF-2α-silenced (shHIF-2α) 1G11 cells (n = 3). The HIF-2α:tubulin densitometry ratio is shown (bottom). **f** VEC mRNA levels, using CD31 as reference, in cells as in **e**. **g** Relative VEC promoter activity in control and shHIF-2α-1G11 cells transfected with mock or SOD3 plasmids. Data shown as mean ± SEM (b-g). *p < 0.05, ***p < 0.001; two-tailed Student's t-test

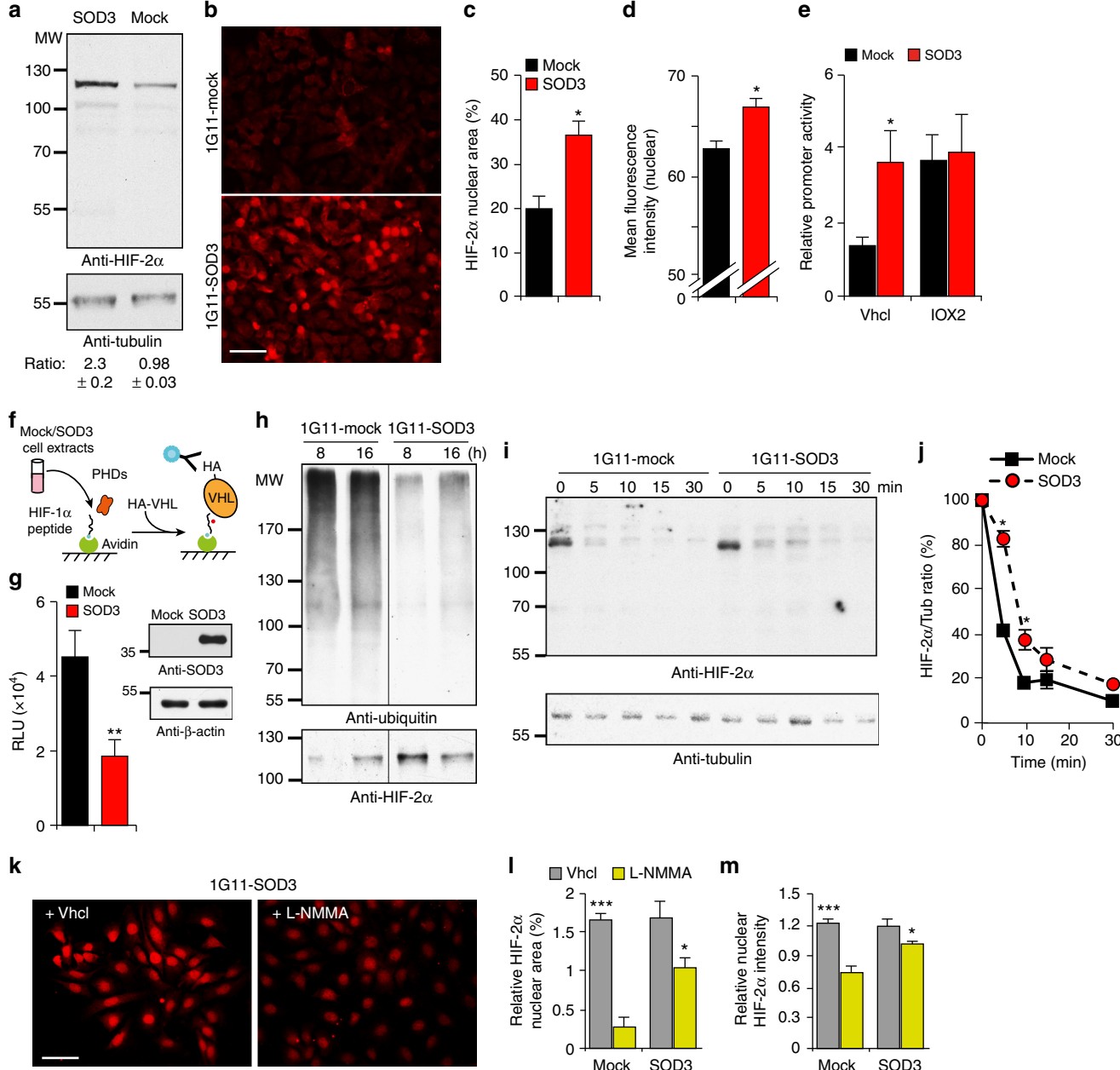

**Fig. 6** SOD3 stabilizes HIF-2α posttranscriptionally via NO. **a** Immunoblot analysis of 1G11-mock and -SOD3 cell extracts with anti-HIF-2α and -tubulin antibodies (n = 2). The ratio between bands is shown (bottom). **b** Immunofluorescence analysis of HIF-2α staining in mock and SOD3-expressing cells (n ≥ 500 cells/condition). **c**, **d** Percentage of nuclear area and mean fluorescence intensity in images in **b**. **e** Relative VEC promoter activity in IOX2- or Vhcl-treated 1G11-mock and -SOD3 cells. **f** Scheme of the ELISA used to detect PHD activity. **g** PHD activity, determined by ELISA, in mock and SOD3-expressing cell extracts. SOD3 levels are shown in these cells as determined by immunoblot (actin loading control; n = 3). **h** Top, HIF-2α ubiquitination analysis in immunoprecipitates of MG132-treated 1G11-mock and -SOD3 cells at the indicated times. Bottom, rehybridization with anti-HIF-2α antibody. **i** Immunoblot of HIF-2α in extracts from 1G11-mock and -SOD3 cells cultured in hypoxia (6 h), cycloheximide treated, and cultured in normoxia for the indicated times. Blots were rehybridized for tubulin as loading control. A representative experiment is shown (n = 4). **j** Quantification of the HIF-2α/tubulin densitometry ratio from **i**, expressed as the percentage of the ratio at time 0 (hypoxic conditions, 100%). **k** Representative images of Vhcl- or L-NMMA-treated 1G11-SOD3 cells stained with anti-HIF-2α antibodies. **l**, **m** Relative percentage of stained nuclear area and mean fluorescence intensity in Vhcl- or L-NMMA-treated 1G11-mock and -SOD3 cells. Values were normalized to the lowest value in the Vhcl-treated group for each cell type (n ≥ 100 cells/condition). For **a**, **c**–**e**, **g**, **j**, **l**, **m**, data shown as mean ± SEM from at least three independent experiments. *p < 0.05, **p < 0.01, ***p < 0.001; two-tailed Student's t-test (**c**, **e**, **g**, **l**, **m**) or two-way ANOVA with Bonferroni post-hoc test (**j**), showing significant differences between cell lines. Bar, 30 μm

It is not obvious how an extracellular enzyme might regulate VEC transcription. SOD3 is not itself a transcription factor but can reach the nucleus[50], as reported for other SODs[51]. In the TMA analyses of CRC patients, individual tumor-associated ECs accumulated intracellular/nuclear SOD3. Nuclear SOD3 protects DNA from oxidative damage[28] and attenuates NF-κB activity by

regulating its oxidative state[52]. In the case of VEC, we propose that SOD3 inhibits PHD activity to prevent HIF-2α ubiquitination and degradation (Supplementary Fig. 12). Since VEC promoter activation is directly proportional to HIF-2α levels (Supplementary Fig. 9c), SOD3-induced HIF-2α stabilization might promote its binding to the promoter and boost VEC

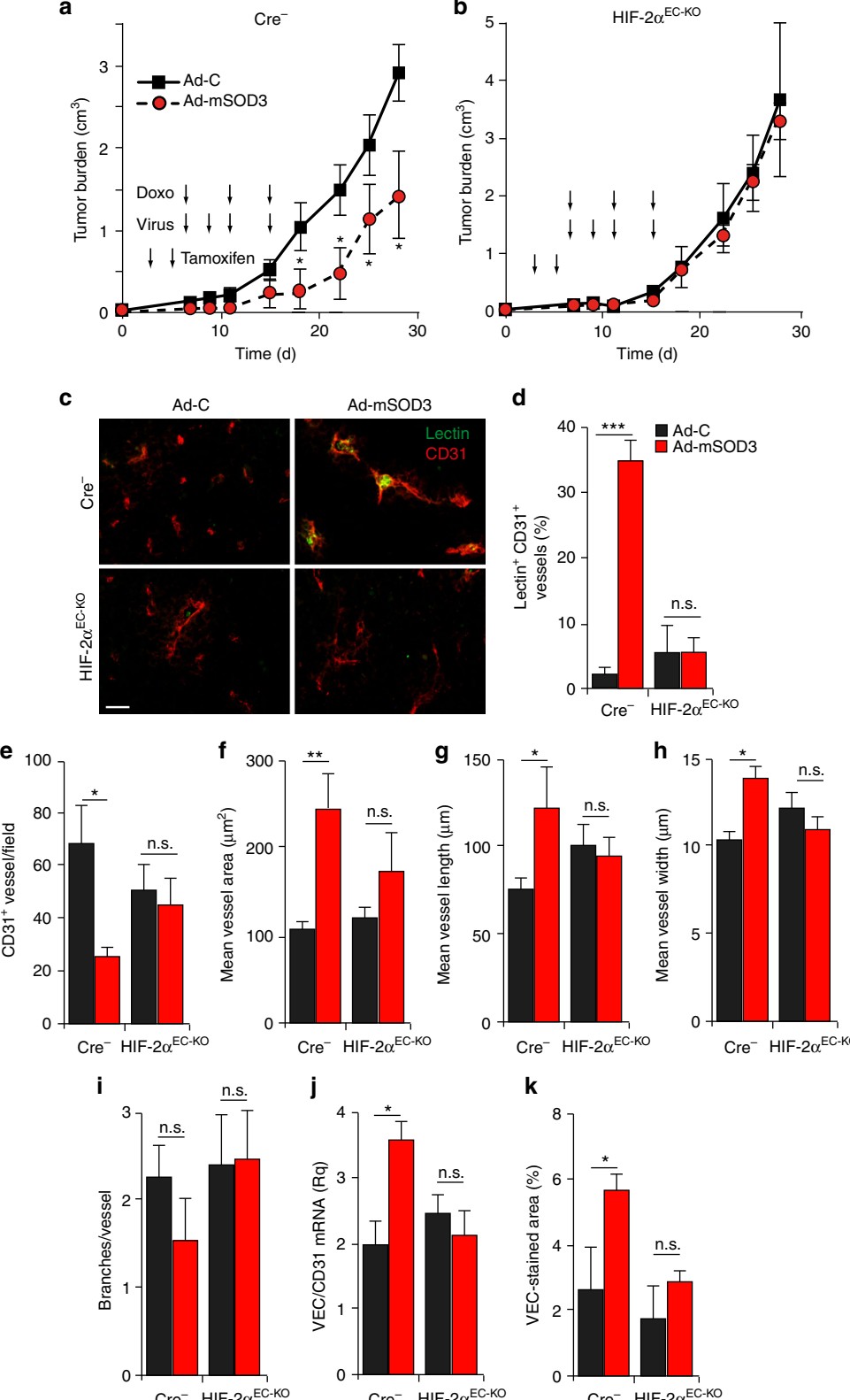

**Fig. 7** Specific HIF-2α ablation in EC abolishes in vivo SOD3 effects. **a**, **b** Growth kinetics of Ad-C- or Ad-mSOD3-injected LLC tumors grafted in control (Cre⁻; **a**) or HIF-2α^EC-KO mice (**b**). **c**, **d** Images of lectin-FITC-perfused (green), CD31-stained (red) vessels in tumors resected at day 18 (≤1 cm³) from control and HIF-2α^EC-KO mice. **e**–**i** Vessel density (**e**), mean area (**f**), length (**g**), diameter (**h**), and branching (**i**) determined by CD31⁺ staining in tumors from **a**, **b** (n ≥ 25 fields/condition). **j** VEC mRNA levels, normalized to those of CD31, in tumors in **a**, **b**. **k** VEC-stained area as a percentage of VEC staining in CD31⁺ structures in images of tumors in **a**, **b** (see also Supplementary Fig. 10c–f); n = 10–20 images/group. *p < 0.05; **p < 0.01; ***p < 0.001; two-tailed Student's t-test. Bar, 50 μm

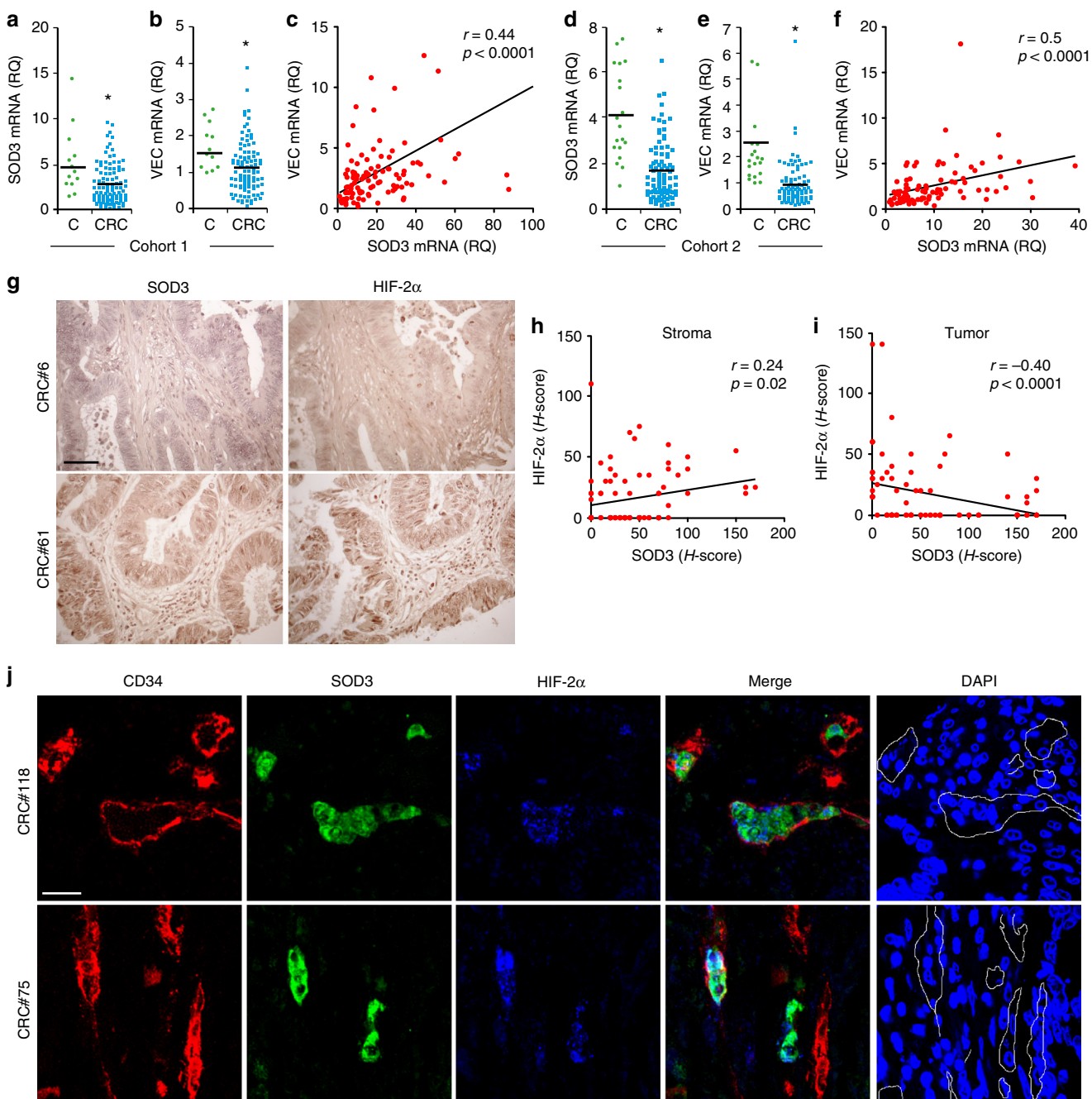

**Fig. 8** Coregulation of SOD3, VEC, and HIF-2α in primary CRC tumors. **a**, **b** Scatter plots showing individual data points and mean for SOD3 (**a**) and VEC (**b**) mRNA levels in non-tumor (C; $n = 13$) and CRC samples (CRC; $n = 102$). **c** Correlation of SOD3 and VEC mRNA levels in the CRC cohort. **d**, **e** Scatter plots as above for SOD3 (**d**) and VEC (**e**) mRNA levels in a validation cohort of 19 non-tumor and 87 stage III CRC samples. **f** Correlation of SOD3 and VEC mRNA levels in the CRC validation cohort. **g** TMA samples stained for SOD3 and HIF-2α, representative of low (#6) and high (#61) SOD3 levels. **h**, **i** Correlation of estimated $H$-scores for SOD3 and HIF-2α in stromal (**h**) and tumor (**i**) compartments. **j** Representative images from two tumor samples stained for CD34 (red), SOD3 (green), and HIF-2α (blue); the three-color merge is shown; nuclei were counterstained with DAPI (see Supplementary Fig. 11 for single-color images). White lines in the DAPI panel contour CD34-stained areas. **c**, **f**, **h**, **i** Spearman's rank correlation test; **a**, **b**, **d**, **e** $*p < 0.05$, two-tailed Student's $t$-test. Bar, 50 μm (**g**) and 20 μm (**j**)

transcription. It was not anticipated that SOD3 would trigger specific HIF-2α binding to the −500/−1000-bp region, in which no HIF-2α binding was detected in mock cells. SOD3-induced HIF-2α binding to the VEC promoter might entail not only HIF-2α stabilization but also local modifications to chromatin structure. Moreover, since HIF-2α but not HIF-1α transactivates the VEC promoter, HIF-2α might cooperate with other transcription partners whose activity or location could be SOD3 influenced.

We propose that SOD3 effects on HIF-2α and VEC expression depend on increased NO availability due to $\cdot O_2^-$ scavenging. This is based on our observation that (i) SOD3 activity and NO synthesis were needed to reduce EC permeability, (ii) NO synthesis blockade reduced and a NO donor increased HIF-2α-mediated VEC promoter activity, and (iii) NO inhibitors prevented SOD3-induced nuclear HIF-2α stabilization.

NO has contrasting concentration- and time-dependent effects on EC barrier function. Lack of eNOS reduces VEGF-induced permeability, suggesting a negative NO role in barrier function[53,54]. Inhibition of endothelial NO production nonetheless accentuates basal and induced alterations in vascular permeability[19,55]. We also found an inverse, dose-dependent DETA-NONOate effect, which increased VEC transcription and reduced EC permeability at low doses, with opposite effects at high doses. In vivo evidence indicates that perivascular NO accumulation reduces tumor vasculature permeability[18], a phenotype similar to that of tumors with high SOD3 levels. The NO reaction with $\cdot O_2^-$ can modify its effect on barrier stability[20]. The NO donor NOC-18 stabilizes HIF-1$\alpha$, whereas the NO/$\cdot O_2^-$ donor SIN-1 promotes its degradation in cerebral ECs; SIN-1 co-addition with SOD stabilizes HIF-1$\alpha$ in ECs[21]. Rather than modifying NO synthesis (eNOS levels are comparable in WT and SOD3$^{-/-}$ EC), SOD3 might maintain a delicate NO/$\cdot O_2^-$ balance that improves EC barrier function and drug delivery.

Although we cannot establish whether the same SOD3/HIF-2$\alpha$/VEC regulatory circuit operates in human malignancies, SOD3 and VEC mRNA levels correlated positively in two independent CRC cohorts. This association is indicative of coregulation between these genes in human CRC. We were unable to confirm this association at the protein level, due to inconsistent results using anti-human VEC antibodies in IHC. SOD3 and HIF-2$\alpha$ nonetheless correlated positively in CRC tumor stroma but were negatively associated in tumor epithelial cells, which suggests that SOD3 stabilizes HIF-2$\alpha$ specifically in stromal but not in neoplastic cells. Most SOD3-stained tumor EC showed HIF-2$\alpha$ staining, whereas unstained ECs lacked HIF-2$\alpha$. If confirmed, the cell type-specific SOD3 effect on HIF-2$\alpha$ levels might be of therapeutic use, given the antithetical HIF-2$\alpha$ function in tumorigenesis; its upregulation in cancer cells promotes proliferation and neoangiogenesis[15,16] and, in ECs, induces vascular normalization and enhanced drug delivery[12,13].

Some tumors showed adjacent SOD3-stained and unstained EC, which appears not to be an artifact but to indicate true stromal heterogeneity. Although SOD3 and NO are diffusible substances that can spread over short distances, extracellular matrix composition and surrounding non-ECs might confine SOD3/NO dispersion, as for other biomolecules[56]. A local inflammatory milieu might also preclude SOD3 expression in specific EC.

In summary, our results identify SOD3 as a regulator of VEC transcription in tumor vasculature by NO-dependent stabilization of HIF-2$\alpha$. This improves perfusion and delivery of chemotherapeutics into murine tumors. SOD3 and VEC are also coregulated in human CRC, although further research is needed to define SOD3 utility as a predictive biomarker for chemotherapy effectiveness.

## Methods

**Plasmids and viral constructs**. SOD3 cDNA was obtained from Open Biosystems and the SOD3$^{S195C}$ mutant was generated by directed mutagenesis (QuickChange Site-Directed Mutagenesis Kit, Stratagene), using the primers described[40]. SOD3 and SOD3$^{S195C}$ were subcloned in the pRVIRES-gfp plasmid and retroviruses produced as described[57]. SOD3 was also subcloned in the shuttle pCMV vector (AdEasy Adenoviral Vector System, Stratagene), and high titer stocks of Ad-mSOD3 and Ad-LacZ were prepared by the Unidad de Producción de Vectores (Centro de Biotecnología Animal y Terapia Génica, Barcelona, Spain). The mouse HIF-2$\alpha$ cDNA cloned in pCMVSPORT-6, the shHIF-2$\alpha$ RNA (pGIPZ-mEPAS1 #481932), and a scrambled shRNA (shNT) were obtained from Dharmacon GE Healthcare. pcDNA3-hHIF1$\alpha$ was provided by Dr. Luis del Peso (Universidad Autónoma de Madrid, Spain). SOD3 was silenced with pLK0.1shRNASOD3 (clone ID NM_011435.2-366s1c1; TRCN0000101354) using pLKO-puro Non-Target (shNT) RNA as control (Sigma-Aldrich).

**Cell lines and overexpression/silencing experiments**. The murine microvascular 1G11 cell line[38] was a gift of A. Mantovani (Humanitas Clinical and

Research Center, Italy) and was cultured on 1% gelatin-coated plates with Dulbecco's modified Eagle's medium (DMEM) containing 20% fetal calf serum (FCS), 2 mM L-glutamine, 1 mM sodium pyruvate, non-essential amino acids, endothelial cell growth supplement (Sigma-Aldrich), heparin, and antibiotics. Their endothelial origin was verified by VEC staining. Stable 1G11-mock and 1G11-SOD3 cells were obtained by retroviral transduction with pRVIRES-gfp and pRV-sod3-IRES-gfp, respectively, followed by cell sorting (MoFlo XDP; Beckman Coulter) using GFP fluorescent emission. Cell lines were analyzed regularly by FACS for GFP expression (Cytomics FC500; Beckman Coulter). SOD3 and HIF-2$\alpha$ were silenced in 1G11 cells with pLK0.1shRNASOD3 (clone ID NM_011435.2-366s1c1; TRCN0000101354) or the shHIF-2$\alpha$ RNA #481932, respectively, using pLKO-puro Non-Target (shNT) RNA as control, with Lipofectamine 3000 (Thermo Fisher Scientific)[58]. Stable cells expressing each shRNA were obtained after puromycin selection (2 µg/ml, Clontech). HIF-1$\alpha$ was silenced by transfecting 1G11 cells with siRNA1 (5′GGGCCGCUCAAUUUAU GAAUAUUAU3′), siRNA2 (5′UACUCA-GAGCUUUGGAUCAAGUUAA3′), or a mixture of both, using Lipofectamine 2000 (Invitrogen); as non-target control, a medium GC content siRNA duplex was used (Stealth RNAi siRNA Negative Control Med GC, 12,935,300, Invitrogen). HIF-1$\alpha$ and VEC mRNA were estimated by RT-qPCR at 48 h posttransfection.

The human dermal microvascular EC line (HDMEC; Cascade Biologics, Portland, Oregon) was used directly from the provider and cultured on 0.5% gelatin-coated plates with MCDB131 medium with 10% FCS, 2 mM L-glutamine, 20 mM HEPES, 10 ng/ml human epidermal growth factor (EGF; PeproTech), 30 µg/ml bovine pituitary extract (Invitrogen), 1 µg/ml hydrocortisone (Sigma-Aldrich), and antibiotics. HDMEC were infected with Ad-hSOD3 (Vector Biolabs) and Ad-LacZ at the indicated m.o.i. BAEC (Clonetics, Lonza) were cultured on 0.1% gelatin-coated plates with low-glucose DMEM medium with 10% FCS and 2 mM L-glutamine. Other cell lines used were obtained from the ATCC and cultured as recommended. All cell lines were tested regularly for mycoplasma contamination. Primary EC from WT and SOD3$^{-/-}$ mice were isolated from the lung and cultured as described[59].

**Human samples**. Freshly frozen stage I–IV tumor samples (cohort 1) were obtained from the Hospital Clínico San Carlos Tumor Bank between 2001 and 2006[43]. We also used formalin-fixed paraffin-embedded tumor samples from stage III CRC patients (cohort 2). For both cohorts, non-tumor samples correspond to colon tissue obtained at >10 cm from the primary tumor in the same patients. These normal colon tissue samples were analyzed histologically by hematoxylin/eosin staining to verify that they did not contain tumor cells; samples in which the pathologist suspected or observed minimal alterations in colon mucosa were discarded. Appropriate informed consent was obtained from all the patients and no personal data were registered. The Hospital Clínico San Carlos Ethical Committee approved this study.

**Animals**. C57BL/6J mice were obtained from The Jackson Laboratory, and SOD3$^{-/-}$ mice have been described[22]. The SOD3 transgenic mice were generated in collaboration with Ozgene Pty. Ltd. (Bentley WA, Australia). The targeting vector was composed of the UbiC promoter, a loxP-flanked STOP cassette, a cassette containing the murine SOD3 cDNA, an IRES sequence, and the GFP cDNA. The entire vector was sequenced to confirm correct assembly. The targeting vector was inserted into the ROSA26 locus by homologous recombination, linearized, and electroporated into C57BL/6J embryonic stem cells. Chimeras were identified and crossed with C57BL/6J mice to obtain the germline-transmitted heterozygous floxed mice (loxP-SOD3KI). loxP-SOD3KI mouse founders were confirmed by DNA genotyping using the primers (fw): 5′GGGAGTGTTGCAATACCTTTCT3′ and (rev): 5′CAGATGACTACCTAT CCTCCCATT3′, which amplify the WT allele, and (fw): 5′CTGAAGCTCCGGTTTTG AACTATG3′ and (rev): 5′GCCTTGAGCCTGGCGAACAGTT3′, which amplify the transgenic allele. Mice with inducible SOD3 expression in EC (SOD3$^{EC-Tg}$) were achieved by breeding two different germ-line loxP-SOD3KI founders with VE-Cad-Cre$^{ERT2}$ mice[37], which has inducible Cre recombinase expression under the control of a modified VEC promoter. Tumorigenesis experiments performed with bi-transgenic mouse lines derived from both loxP-SOD3KI founders yielded similar results. Inducible deletion of HIF-2$\alpha$ in EC (HIF-2$\alpha^{EC-KO}$) was achieved by crossing HIF-2$\alpha^{f/f}$[42] with VE-Cad-Cre$^{ERT2}$ mice. In all experiments involving SOD3$^{EC-Tg}$ and HIF-2$\alpha^{EC-KO}$ mice, Cre$^+$ and Cre$^-$ littermates were used for tumorigenesis studies. Mouse experiments were approved by the Comunidad de Madrid (PROEX 399/15) and the CNB Ethics Committees and carried out with 2–5-month-old female mice in strict accordance with the Spanish and European Union laws and regulations concerning care and use of laboratory animals.

**Syngeneic tumor models**. Adherent growing LLC (>98% live cells) were harvested, and single-cell suspensions of $5 \times 10^5$ cells in 100 µl phosphate-buffered saline (PBS) were injected subcutaneously into the right flank of the syngeneic mouse lines indicated above; only female mice were used. Tumors were measured periodically with calipers and tumor volume was calculated using the formula $V =$ width$^2 \times$ length/2. When tumors reached an average size of 100 mm$^3$, LLC tumor-bearing mice were randomized for treatment by a technician blind to treatments. In tumor experiments involving use of statins, mice received one daily intraperitoneal

(i.p.) injection of Vhcl (5% ethanol) or Lov (15 mg/kg; Sigma-Aldrich) starting at day 7 after LLC inoculation. In tumor experiments involving use of SOD3$^{EC-Tg}$ and HIF-2α$^{EC-KO}$ mice, tamoxifen (Sigma-Aldrich) was diluted in ethanol and corn oil, heated at 100 °C and injected (1 mg/mouse, i.p.) on days 3 and 6 post-LLC inoculation. Tamoxifen-induced SOD3 overexpression in EC was determined by immunochemical staining of SOD3 in tumor sections; tamoxifen-induced excision of HIF-2α was determined by RT-qPCR in purified EC from tumors grafted in Cre$^+$ and Cre$^-$ mice. In tumor experiments involving the use of adenovirus (SOD3$^{-/-}$ and HIF-2α$^{EC-KO}$ mice), Ad-mSOD3 or Ad-C viruses (10$^9$ pfu/50 µl, intrathecal) were injected on days 7, 9, 11, and 15 after LLC inoculation. For all tumor models, Vhcl or Doxo (2.5 mg/kg; Farmitalia Carlo Erba, Italy) was administered i.p. on days 7, 11, and 15 postimplantation. Tumor volume was measured blind to treatment or genotype.

For tumor cell fractionation, we used LLC-GFP cells generated by retroviral transduction of LLC cells with pRV-IRES-GFP; transduced cells were selected by cell sorting (MoFlo XDP; Beckman Coulter) using GFP fluorescent emission. LLC-GFP cell in vitro proliferation capacity was similar to parental LLC cells. Tumors were generated by subcutaneous injection of LLC-GFP cells and mice were treated with Vhcl or Lov as above; at day 21, tumors, lungs and spleen were excised. Tumors were digested (16 h, 37 °C) in DMEM/F12 (1:1) with collagenase (300 U/ml)/hyaluronidase (100 U/ml) (Stem Cell Technologies) and DNase I (Roche). Digestion was terminated by addition of HBSS (Gibco) with 10 mM HEPES and 2% FCS[60]. The resulting single-cell suspensions were stained with anti-CD45-APC (clone 30-F11, eBioscience) and anti-CD31-PE antibodies (clone MEC13.3, BD Pharmingen); EC (CD31$^+$), hematopoietic (CD45$^+$) and LLC-GFP cells were isolated by cell sorting (see below). Isolated cells were used for SOD3 mRNA quantification (see below). ECs were isolated from the lungs of these mice, as described[59]. After erythrocyte lysis, hematopoietic cells from mechanically disrupted spleens of tumor-bearing mice were isolated by positive selection of CD45$^+$ cells with sheep anti-rat magnetic beads (Invitrogen) precoated with the anti-CD45 antibody. Lungs and spleen showed no macroscopic or microscopic signs of metastasis.

**Doxo quantification**. Intratumor Doxo levels were quantified blindly in tumor extracts 1 day after the last Doxo dose, using high-performance liquid chromatography coupled to fluorescence detection, with daunorubicin as internal standard[33].

**qPCR of human and mouse genes**. Total RNA was extracted from mouse tumors, isolated cells from tumors and healthy organs, and cell lines, using Tri-Reagent (Sigma Aldrich) and RNEasy Mini or Micro Kit (Qiagen). RNA from frozen human samples was extracted using TRIZOL (Invitrogen, Carlsbad, CA) and from human formalin-fixed paraffin-embedded tumor samples using the RNeasy FFPE Kit (Qiagen). RNA was treated with DNAse (RNeasy Microkit, Qiagen) and quantified with a NanoDrop ND-1000 (Thermo Scientific). RNA quality was determined with an Agilent Bioanalyzer 2100 (Agilent Technologies). First cDNA strand was synthesized from 0.2–2 µg total RNA (High-capacity cDNA Archive Kit, Applied Biosystems), using random primers. mRNA levels of SOD3, CD31, VEC, DLL4, eNOS, FLT1, ICAM1, VCAM1, ROBO4, TIE2, VEGF-A, NOTCH1, NOTCH2, NOTCH4, HIF-1α, and HIF-2α were quantified in an ABI PRISM 7900HT System (Applied Biosystems), using a SYBR Green-based reaction mix (FluoCycle; EuroClone), with the primers listed in Supplementary Table 1. β-Actin (mouse samples) and *RPL10A* genes (human samples) were used for normalization except for VEC mRNA analyses, which were normalized with CD31 mRNA levels. Values for each gene are expressed as relative quantity (Rq), calculated as 2$^{-\Delta\Delta Ct}$ relative to the sample with the lowest expression.

**SOD3 induction in cultured cells**. For in vitro experiments, Lov was activated as described[61]. 3T3 cells were serum depleted (24 h) before incubation with TNF-α (10 ng/ml; Peprotech) and various doses of activated Lov, simvastatin, or atorvastatin (both 1 µM; Calbiochem) or their vehicles. Media was refreshed every 24 h with TNF-α and statins. Cells were lysed after 48 h and RNA was extracted as above.

**Immunoblot**. Tumor and cell extracts were prepared with RIPA buffer, proteins resolved by sodium dodecyl sulfate-polyacrylamide gel electrophoresis (SDS-PAGE) and immunoblotted with rabbit anti-SOD3, rabbit anti-VEC, mouse anti-pan-cadherin (H-19, Sigma Aldrich), rabbit anti-HIF-2α (NB100-122, Novus Biologicals, or ab109616, Abcam), rabbit anti-HIF-1α (PA1-16601, Thermo Fisher Scientific), or rabbit anti-PHD2 (#4835, Cell Signaling); mouse anti-β-actin, -tubulin (Sigma Aldrich), or -GAPDH (clone 6C5, Abcam) were used for loading controls. Mouse lung or kidney extracts were used as positive controls for SOD3. For HIF-2α ubiquitination analyses, 1G11-mock and -SOD3 cells in exponential growth were treated with the proteasome inhibitor MG132 (1 µM; S2619, Selleckchem) for 8 or 16 h. Cells were lysed in RIPA buffer supplemented with 1% SDS, 20 µM NEM (*N*-ethylmaleimide; Sigma-Aldrich), and a complete Mini Protease Inhibitor Cocktail (Roche); total lysates (250 µg) were immunoprecipitated with 2 µg rabbit anti-HIF-2α antibody (sc 28076, Santa Cruz Biotechnology) and protein A/G magnetic beads (Protein A/G MagBeads, GenScript). Bound proteins were eluted with 0.1 M glycine, pH 2.3, then neutralized, resolved by SDS-PAGE,

and immunoblotted sequentially with mouse anti-ubiquitin (clone P4D1; sc-8017, Santa Cruz Biotechnology) and rabbit anti-HIF-2α (ab109616, Abcam) antibodies.

For cycloheximide chase assays, 1G11-mock and -SOD3 cells were cultured in hypoxia (3% O$_2$, 37 °C, 6 h) and then treated with cycloheximide (50 µg/ml, 10 min; Sigma Aldrich) in hypoxia. Cells were then shifted to normoxia (21% O$_2$) and lysed at the indicated times using RIPA buffer supplemented with 1% SDS and a complete Mini Protease Inhibitor Cocktail (Roche). Resolved proteins were immunoblotted with anti-HIF-1α, -HIF-2α, and -tubulin antibodies; the HIF-1/2α/tubulin ratio was calculated and expressed as a percentage relative to time 0.

Densitometric measurements for all immunoblots were performed with the ImageJ software (NIH). Full immunoblots for main and supplementary figures are shown in Supplementary Figs. 13 and 14, respectively

**Immunohistochemistry**. All methods for IHC and IF of mouse tumors have been described[33,62]. Briefly, tumors were excised at days 16–18 after tumor cell inoculation and snap-frozen in Tissue Freezing Medium (OCT; Jung) or fixed with 10% neutral buffered formalin (3 h) and paraffin embedded. Sections (5–20 µm) were prepared and used directly (frozen tissue) or deparaffinized, rehydrated, and subjected to heat-induced epitope retrieval in citrate buffer (0.01 M sodium citrate pH 6; 20 min). For immunostaining, the following rabbit antibodies were used: anti-CD31 (MEC13.3, BD Bioscience), -SOD3 (Cloud Clone), -VEC (LS-B2138, LSBio), -HIF-2α (PA1-32216, Thermo Fisher Scientific), and -3-NT (#06–284, Cell Signaling). Sections were incubated with appropriate fluorescently conjugated secondary antibodies (Alexa 488 or 546, Molecular Probes) or with peroxidase-labeled anti-rabbit IgG (Dako), followed by amino ethyl carbazol (AEC; Enzo) and hematoxylin counterstaining. Sections were mounted with 4,6-diamidino-2-phenylindole (DAPI)-containing Fluoromount-G (SouthernBiotech; fluorescence analyses) or Dako Faramount Aqueous Mounting Media (Dako; conventional IHC) and analyzed with an Olympus FluoView 1000 confocal microscope with a ×60 1.4 oil plan-Apo objective or with a Leica (DM RB) microscope equipped with an Olympus DP70 camera.

**TMA analyses**. Microarrays were prepared from hCRC samples[43] and IHC was performed in deparaffinized samples after antigen retrieval in citrate buffer (pH 6.0, 20 min) by incubation with goat anti-SOD3 (AF3420, R&D Systems), rabbit anti-HIF-2α (NB100-122, Novus Biologicals), or mouse anti-CD34 antibodies (QBEnd-10, M7165 Dako), followed by appropriate peroxidase-labeled secondary antibodies. The reaction was developed with AEC or diaminobenzidine (CD34) as chromogens, and hematoxylin counterstained. In all cases, sections from normal colon mucosa distant from the tumor site were used as controls. Staining was evaluated in a Leica DM500 optical microscope by a single pathologist (M.J.F.-A.) blinded to experimental data. The percentage of stained cells and staining intensity (scored as 1–3) were recorded for each gene, and the *H*-score was calculated as the product of these parameters[63] for the epithelial tumor cell compartment and for tumor-associated stromal cells. In IF analyses, samples were incubated with all three antibodies, followed by appropriate secondary antibodies. Images were captured on an Olympus FluoView 1000 confocal microscope.

**HIF-1/2α staining and nuclear quantification**. Immunofluorescence analyses of cultured EC were performed in 1% gelatin-coated Nunc Lab-TEK chamber slides. Untreated or L-NMMA (100 µM; Cayman)-treated cells were fixed with 4% paraformaldehyde (PFA; 10 min, 20 °C) for HIF-1α staining or with ethanol:acetic acid (95:5, 1 min, −20 °C) for HIF-2α staining. Cells were then permeabilized with 0.1–0.3% Triton X-100 (5 min, 20 °C) and stained with rabbit anti-HIF-1α (PA1-16601) or -HIF-2α (PA1-32216; Thermo for 1G11 cells; NB100-122; Novus Biologicals for HDMEC cells), followed by Cy3-labeled goat anti-rabbit antibody; nuclei were counterstained with DAPI Fluoromount-G (Southern Biotech). As positive control for HIF-1α, 1G11 cells were treated with CoCl$_2$ (100 µM, 6 h, 37 °C) before fixing. Images were acquired with a Leica DMI6000B microscope equipped with a Hamamatsu camera ORCA-R2 (HIF-2α). An open source MATLAB-based computational platform was developed for automatic quantification of HIF-2α-associated fluorescence after background subtraction. Nuclei were segmented using DAPI images and the segmentation applied to the Alexa546 channel to measure HIF-2α-stained pixels within the nuclear area; the ratio of nuclear staining to total nuclear area was then calculated. Reactive oxygen species levels in 1G11-mock and 1G11-SOD3 cells, alone or in tumor-conditioned medium, were determined by staining of the dihydroethidium probe (2 µM, 30 min, 37 °C; Invitrogen).

**Tumor perfusion**. Tumor-bearing mice (days 18–21 post-LLC inoculation) received injections (intravenous (i.v.)) of FITC-lectin (100 µg; Vector Laboratories) and were heart-perfused with 10% neutral buffered formalin. Tumors were excised, snap-frozen in OCT, and 50 µm-thick sections were analyzed with a Radiance 2100 confocal system (BioRad) on a Axiovent 200 microscope (Zeiss; excitation/detection 488/500–560 nm). In some experiments, vessels were 3D-reconstructed from section stacks (1 µm step size) with Imaris v7.3.1 software (Bitplane). The number of double (CD31 and lectin)- and single (CD31)-stained structures was determined in tumor sections using ImageJ software. RGB images were transformed to 16–bit and, after threshold adjustment, the Analyze Particle tool was applied using settings

for particle size (>10 microns) and circularity (0–1). Perfusion was calculated as the percentage of double-positive (CD31$^+$lectin$^+$) to CD31$^+$ structures.

**CD31 staining and quantification of blood vessel parameters**. End-point tumors were snap-frozen in OCT and 20 µm sections were stained sequentially with anti-CD31 and fluorescently-labeled secondary antibodies. The number of CD31-stained structures and average size were determined with Image J as above. To determine other vessel parameters, we used histogram equalization to increase contrast and allow detection of small and lightly stained structures. A Hessian-based Frangi filter was then applied to enhance vessel fibrillar structure[64]. A vessel mask was obtained by applying Otsu thresholding; this mask was used to visualize vessel skeleton using morphological thinning operations to calculate length and branch points of each vessel. Euclidean distance transform from the vessel mask to the edge of segmentation was also performed. Vessel diameter was calculated by the values of the Euclidean distance transform lying on the skeleton (central values). All calculations were performed using MATLAB Release 2017a (The MathWorks). For Vhcl- and Lov-treated tumors, lectin-perfused images (see above) were used for analyses.

SEM was performed as reported[33].

**VEC staining and quantification**. Tumor-bearing mice (day 17–20 post-LLC inoculation) were perfused with 2% PFA in PBS, fixed in PFA, and included in OCT. Tumor sections were stained with anti-VEC and -CD31 antibodies, followed by appropriate labeled secondary antibodies, and analyzed in an Olympus Fluo-View 1000 confocal microscope with a x60 1.4 oil plan-Apo objective. Images were transformed to 8-bit with ImageJ and, after threshold adjustment, the Image calculator tool was used to select CD31- and VEC-stained coincident regions. The ratio between VEC- and CD31-stained areas was calculated and expressed as a percentage.

For quantification of VEC-stained length in tumor vessels, images were segmented based on CD31 staining using the Otsu method; this allowed the selection of blood vessels. Hysteresis thresholding was applied to VEC staining in the CD31 area. Morphological thinning was run to measure the length of VEC-stained structures using MATLAB.

For VEC staining in human and mouse EC, cells were plated on 1% gelatin-coated Lab-TEK chamber slides (Nunc), fixed with 4% PFA (10 min, 20 °C), permeabilized with 0.25-0.3% Triton X-100 (5 min, 20 °C) and stained with appropriate antibodies for mouse or human VEC (see above). In some experiments, 1G11-mock and -SOD3 cells were pretreated with VEGF$_{165}$ (50 ng/ml, 2 h; Peprotech 450–32) before staining. Images were acquired with the Olympus FluoView 1000 system. Junctional length and gap index were quantified as described[39], with a minimum of 5 fields/condition.

**Permeability assays**. In vivo vessel leakage was analyzed in tumor-bearing mice (day 17) by i.v. co-injection of Texas Red-conjugated dextran 70 kDa (250 µg; Molecular Probes) and FITC-lectin (100 µg). After perfusion with 2% PFA in PBS, tumors were fixed (2% PFA, 3 h) and snap-frozen in Optimal Cutting Temperature (OCT). Tumor sections were analyzed by confocal microscopy (Olympus FV1000), and images were quantified using the confocal software (FV10-ASW, Olympus). Vessel leakage is the ratio between fluorescence intensity of extravasated dextran and perfused vessel counts.

For in vitro permeability analyses, $2 \times 10^5$ 1G11 or HDMEC (stable, transfected with indicated plasmids, or infected with Ad-hSOD3 or control) were seeded on 1% gelatin-coated transwells (24 mm diameter; 0.4 µm pore; Costar); this cell concentration formed confluent monolayers after 24 h culture. Cell monolayers were incubated with tumor cell-conditioned medium, and EC permeability was assayed using FITC-dextran (40 kDa; 1 mg/ml) in phenol red-free DMEM. When indicated, the pan-NOS inhibitor L-NMMA, different doses of the NO donor DETA-NONOate or VEGF$_{165}$ were added to cells before assay. Permeability was determined by fluorescent readout in the lower chamber (FilterMax F5 Multi-mode Reader, Molecular Devices). Fluorescence values were extrapolated to a standard curve constructed using known amounts of FITC-dextran.

**NO detection**. For NO detection in vivo, Vhcl- and Lov-treated tumor-bearing mice (day 18) were co-injected with FITC-lectin and the NO-sensitive fluorescent probe DAR-1 (320 µg in dimethyl sulfoxide (DMSO); Sigma-Aldrich). After 1 h, mice were perfused by intracardiac injection of PBS, and tumors were excised, fixed with 2% PFA (3 h), and snap-frozen in OCT. Thick sections (40 µm) were analyzed by confocal microscopy (Leica TSC SP5) and quantified with ImageJ; pseudocolor images were obtained with the Leica confocal software (LAS AF 2.6.0).

Quantification of intracellular NO in EC was as described[65]. Briefly, 1G11 cells were transduced with Ad-C and Ad-mSOD3 viruses and were incubated the next day (3 h, 37 °C) with the NO-sensitive probe 4,5-diamofluorescein (DAF2, 10 µM; Sigma-Aldrich). Cells were analyzed by FACS, using DAF2-untreated cells as negative control.

**Promoter assays**. pGL3-VEC-Luc (2.4 kb VEC promoter)[66] and p9xHIF-Luc[67] plasmids were provided by Q. Xu (King's College London British Heart Foundation Centre) and L. del Peso (Universidad Autónoma de Madrid), respectively. Serial 5′-

deletion mutants of the VEC promoter were generated by PCR using pGL3-VEC-Luc as template. HRE sites in the VEC promoter were mutated by replacing the CGT sequence in the HRE consensus site[68] with AAA or TTT, using the following oligonucleotides for site-directed mutagenesis (HRE1: 5′GCTGAAAGTGATT GTCTGTCTTTTTTCTCAGCTGCCCG3′; HRE2: 5′CCCTGGTTGGTCCATGGT CAAAAGAAGCCCATCACCCAG3′, and HRE3: 5′GGATTAAAGGAAAGCGC CACCTTTCCTGGCTGAATG3′).

For the reporter assays, 1G11 or 3T3 fibroblast cells were transiently transfected using Lipofectamine 3000 (Thermo Fisher Scientific) with a mixture of plasmids (500 ng total DNA) containing pGL3-VEC-Luc, pGL3-VEC-Luc mutants or p9xHIF-Luc plus pRVIRES-gfp (mock control), pRVSOD3-IRES-gfp, pcDNA3-hHIF1α, or pCMVSPORT6-HIF-2α, and pRL-SV40-Luc Renilla as internal control, at a 6:3:1 ratio (for 1G11) or 4:8:1 (for 3T3). In experiments involving different HIF-1α and HIF-2α doses, the transfected DNA amount was kept constant with an empty pcDNA3 plasmid. In some experiments, transfected cells were incubated with DETA-NONOate, L-NMMA, or IOX2 (Cayman), using an equivalent DMSO concentration as vehicle. Firefly and Renilla luciferase activities were measured 72 h posttransfection using the Dual-Luciferase Reporter Assay System (Promega). Relative promoter activity was defined as the ratio of firefly activity to Renilla activity, with that of control set as 1.0.

**ChIP assay**. ChIP assay was performed with the EZ-ChIP Kit (Millipore) according to the protocol provided. Briefly, 1G11-mock and 1G11-SOD3 cells were fixed (10 min, room temperature (RT)) with 1% PFA in culture medium and quenched with glycine (5 min, RT). After medium removal, $2 \times 10^7$ cells/ml were harvested in lysis buffer (4 °C, 15 min), and chromatin was sheared by sonication (20 cycles of 30 s on and 30 s off; Bioruptor Pico, Diagenode) in aliquots of 0.2 ml. As input reference, 1% of each cell lysate was stored; the remainder was immunoprecipitated (4 °C, 14 h with rotation) with rabbit anti-HIF-2α (NB100-122; 20 µg) or goat anti-KLF-4 antibody (sc12538, Santa Cruz Biotechnology; 10 µg) and purified rabbit or goat IgG as respective controls. Immune complexes were captured using Protein G agarose beads, pelleted by centrifugation, and washed with low-salt buffer, followed by high-salt buffer and Tris-EDTA buffer (25 mM Tris-HCl, 150 mM NaCl, 1 mM EDTA, pH 7.2) to minimize nonspecific binding. Chromatin immunocomplexes were eluted from beads with elution buffer (100 mM NaHCO$_3$, 1% SDS), and the protein/DNA link was reversed by incubation (4 h, 65 °C) with 5 M NaCl, followed by proteinase K (10 µg/µl, 2 h, 45 °C). DNA was then purified using spin columns, and VEC gene promoter sequences were analyzed by RT-qPCR with the primers ChIP1-(fw)5′GAAGTGCTACCCTGGCA-GACGTG3′, (rev) 5′TCCATAGCATTCAACTACTGCGTG3′ (amplicon: 236 bp), and ChIP2-(fw) 5′TTTGCTGACTCAGACCTATGGCTA3′, (rev)5′GGGCAGCT-GAGAAACGTGACAGAC3′ (amplicon: 225 bp). Dissociation curves showed that PCR yielded single products.

The relative quantity of amplified product in the input and ChIP samples was determined as described[69]. Briefly, serial dilutions of the pGL3-VEC-Luc were amplified and a standard curve was drawn in log2 base, plotting CT values (x axis) vs. the corresponding dilutions (y axis). The CT values determined in ChIP and input samples were calculated using this curve, and the value obtained was used to calculate the power of two. The signal relative to input was calculated as the fraction of ChIP to the input, after subtraction of the value obtained from IgG ChIP. Data shown are ChIP results obtained from at least four independent replicates.

**Hydroxylase activity assay**. Endogenous PHD activity was measured in mock- and SOD3-transfected cells using HIF-1α as substrate[70]. Briefly, cells were transiently transfected with pRVSOD3-IRES-gfp or pRVIRES-gfp using the calcium phosphate method; transfection efficiency was 70–90%, as analyzed by FACS using GFP as marker. Cell extracts were prepared at 48 h posttransfection using a hypotonic buffer (20 mM Tris-HCl pH 7.5, 5 mM KCl, 1.5 mM MgCl$_2$), mechanical disruption with a Dounce homogenizer, and passage through a 30-g needle. Soluble material was obtained by centrifugation (12,000 × g, 30 min, 4 °C). The PHD activity of these extracts was analyzed by ELISA.

High-binding white 96-well plates were coated (overnight, 4 °C) with ExtrAvidin (5 µg/ml; 100 µl/well; Sigma-Aldrich) in coating buffer (0.1 M carbonate buffer pH 9.4). All other incubations were carried out at RT. After blocking with 2% bovine serum albumin in PBS–0.1% Tween 20 (2 h), plates were incubated (1 h) with 200 ng/well biotinylated HIF-1α peptide (DLDLEMLAPYIPMDDDFQL; CNB Proteomics Service, Madrid, Spain); a hydroxylated version of the peptide (DLDLEMLAP(OH)YIPMDDDFQL) was used for specificity control assays. After three washing steps with PBS–0.1% Tween20 and one with hypotonic buffer, 200 µg/well of the cell extracts were used in a hydroxylation reaction in hypotonic buffer (with 0.5 mM α-ketoglutaric acid, 0.1 mM FeCl$_2$, and 0.5 mM Na-L-ascorbate) (2 h). After five washing steps, 5 µl/well in vitro-transcribed and translated HA-VHL protein (HA-VHL-pRc/CMVm Addgene, plasmid 19,999) was added (2 h), followed by incubation (1 h) with a peroxidase-conjugated mouse anti-HA antibody (clone HA-7; Sigma-Aldrich). Peroxidase activity bound to the solid phase was detected with the Super-Signal ELISA Pico chemiluminescent substrate (50 µl/well; Thermo Fisher Scientific), and light emission was measured (200 ms) in an Infinite M200 microplate reader (Tecan). Relative luminescence units are mean ± SEM of triplicate samples.

**Statistical analysis**. For in vitro experiments, data are shown as mean ± SEM of at least three independent experiments with at least three technical replicates. For in vivo experiments, data are shown as mean ± SEM of individual mice pooled from at least two independent experiments. Sample size in animal studies was estimated using the power method, and values corrected for 20% attrition. For staining of in vivo material, at least five mice were used. For data with normal distribution and homogeneity of variances, statistical significance was calculated with a two-tailed Student's $t$-test for comparison of two independent groups and one-way or two-way ANOVA with Dunnett's, Tukey's, or Bonferroni's post-hoc test for multiple comparisons. When the above requirements were not fulfilled, data were analyzed using non-parametric tests. Correlation analyses were performed using the non-parametric Spearman's rank correlation coefficient. Differences were considered significant when $p < 0.05$. All statistical analyses were performed using the Prism software.

**Data availability**. The authors confirm that all relevant data and materials supporting the findings of this study are available on reasonable request. This excludes materials obtained from other researchers, who must provide their consent for transfer.

**Code availability**. MATLAB programs generated for HIF-2α image quantification can be accessed on reasonable request.

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

## Acknowledgements

The authors thank A González-Martín for critical reading; L Iruela-Arispe, A Mantovani, Q Xu, L del Peso, and LM Redondo for reagents; MC Moreno and S Escudero for help with cell sorting; RM Peregil, N Rodríguez, N Dalmau, and JM Ballestero for technical assistance; and C Mark for editorial support. This work was funded by grants from the Spanish MINECO (SAF2014-54475-R, SAF2017-83732-R; AEI/FEDER, EU), the Comunidad de Madrid (B2017/BMD-3733; Inmunothercan-CM), and the Domingo-Martínez Foundation to S.M. S.M.-P. is supported by a Miguel Servet contract (CP09/00100), L.C.-R. is funded by the FPU program, and I.H.-M. and D.M.-R. were AECC summer fellowship recipients.

## Author contributions

E.M. and S.M. conceived the study. E.D.-R., B.P.-V., M.P.-C., T.O., and S.M.-P. provided samples and reagents. J.C. and G.F. determined Doxo. P.M.-G. developed MATLAB programs and quantified HIF-2α images. M.J.F.-A. quantified T.M.A. E.M., L.C.-R, M.J.F.-A., D.M.-R., M.T., I.H.-M., and R.A.L. performed most of the experiments. S.M. wrote the manuscript. All authors read and discussed the manuscript.

## Additional information

**Competing interests:** The authors declare no competing financial interests.

