## [Peer Review File · Nature Communications]

Reviewers' comments:

Reviewer #1 (Remarks to the Author):

The ability of statins to inhibit different cancer types has been widely reported. In the present study, authors identified the ability of lovastatin to induce SOD3 with concomitant effect on tumor inhibition via altering HIF2a and Vascular Endothelial Cadherin (VEC) expression. Authors demonstrate the effect of lovastatin on the delivery of doxorubicin (DOXO) in tumors, enabling better tumor response to the drug.

Mechanistically, authors demonstrate that Lovastatin (LOV) mediates this effect via increasing SOD3 reducing NO oxidation preventing PHD3 modification of HIF2a, resulting in elevated HIF2a stability / expression. In turn, HIF2a activates VEC transcription which improve endothelial barrier function, normalizes the vasculature resulting in improved transport of small molecules – such as doxorubicin – into the tumor. Overall, authors provide both genetic and pharmacological support for their model, providing an interesting and potentially important insight into mechanism underlying statin's effect on tumor response. Notwithstanding, the following points deserve attention.

Major

What is the mechanism underlying LOV increase of SOD3 expression?

Authors demonstrate that the increase in DOXO effect in tumors is lost in mice lacking SOD3. Where should SOD3 be expressed to impact the tumor? In stroma / epithelial cells or in the tumor. Along these lines, would loss of SOD3 in tumors be sufficient to attenuate DOXO effect? Is it sufficient to reduce SOD3 expression in the stroma? Authors need to establish decisively which are the tissue types that exhibit the change in SOD3 and later HIF2a expression.

Is the effect of LOV on SOD3 dose dependent? Likewise, are the changes in HIF2a and VEC expression levels (does dependent) corresponds to the level of LOV?

Would other statins equally affect SOD3? Would induced expression of HIF2a, on its own, phenocopy the changes seen upon LOV treatment?

What is the role of HIF1a in this process, and how does HIF2a being selectively activated? SOD3 was previously reported to induce HIF1a, was this possibility ruled out?

Most if not all studies were carried out in the 1G11 cell line, which does not appear to express endogenous SOD3. What is the level of SOD3 in other cell lines (i.e. HMEC-1)? Both overexpression and inactivation of SOD3 in different cell lines is required.

Additional points:

Fig 1b: 3- NT panel staining in SOD3^{-/-} after LOV treatment is high, inconsistent with the data shown in panel 1a.

Fig 2a: decrease in FITC-Lectin in SOD3^{-/-} is also seen in control. Sup data also reveal increase FITC lectin in SOD3^{-/-} after Lov. Please explain.

Fig 2g and elsewhere: data shown need to be quantified and statistically calculated.

Fig 2k: What is the level of VEC mRNA in SOD3^{-/-} mice?

Fig 3c: although tumors are smaller in transgenic mice expressing SOD3 degree of inhibition varies

between the experiments. For example, the tumor size in SOD3EC-Tg shown here is equal to the size shown in the treated groups shown in 1d groups.

Fig 4g: statistical analysis need to be included

Fig 4k: the changes, seen only in one concentration, is hard to understand.

Fig 5c: the differences shown between CHIP 1 and CHIP 2 per Hif2a expression are not clear. What is the evidence for specific HIF2 binding? Is the binding significantly greater? What is the level of HIF1a, which is subject to PHD regulation? Would PHD inhibitors, available commercially, phenocopy the effect of LOV?

Fig 6h: degree of ubiquitination claimed for is not compelling. Data per the half-life of HIF2a as well as HIF1a is needed.

Fig 6i: assessment of HIF1a is needed

Fig 7c-h: what are the changes upon inducible HIF2a vs. HIF1a expression? Inclusion of the PHD inhibitor?

Fig 8g, j: quantification with respective statistical analysis is required.

Minor points:

More detailed legends are needed throughout the ms; often it is hard to understand what is being shown.

Reviewer #2 (Remarks to the Author):

The study by Mira et al describes the role of SOD3 as a regulator of vascular permeability in a tumor setting. Via NO, SOD3 reduced HIF PHD activity, thereby promoting HIF-2 α stability and binding to HRE motifs in the VE-cadherin (VEC) promoter. The authors exploit a range of genetic models and pharmacological inhibitors to show that Lewis Lung Carcinoma (LLC) treatment with Doxorubicin is more efficient with high SOD3 levels, due to stabilized endothelial junctions. They state that their data define a role for SOD3 as a key regulator of tumor endothelium structure and function due to VEC upregulation. However, still the study is incomplete and key data for their main conclusions are missing or, of insufficient quality.

1. One main question is whether SOD3 regulates tumor endothelium alone or also normal endothelium. This distinction between normal and tumor endothelium is implied but not addressed. As SOD3 is secreted it would also be surprising if the effects were so compartmentalized. It would have been much preferable (and easier to read) if the authors had started their study with characterizing the phenotype of the Sod3^{-/-} mouse.

2. In the absence of the Lovastatin treatment, there is no difference in the expression level of SOD3 in wt and SOD3^{-/-} mice (Fig. 1a). This is confusing. Is it tumor cell-derived SOD3 that is measured? The authors could do a better job here in distinguishing different pools of SOD3 in the tumors. The PCR has to be quantitative. See also below comment 7. One gets doubtful whether SOD3 is expressed in endothelial cells at all?

3. The LLC tumors seem to respond better to Doxo in the presence of SOD3 but overall, SOD3-positive effects are dependent on Lovastatin cotreatment. Lova upregulates SOD3 but it must have a variety of other effects. This should be discussed.

4. The authors claim that SOD3 mediates increased VEC and stabilized junctions. Nowhere do the authors show the morphology of junctions with sufficiently high resolution whether in the absence or presence of SOD3, in normal vessels or in the tumor vasculature. Please show and quantify upregulation of VEC immunostaining in vivo in WT and Sod3^{-/-} mice and when SOD3 is overexpressed in ECs.

5. The LLC tumors are grown too large. In some cases, they reach 5 cm³, which is not acceptable ethically, moreover, how do the authors quantify such large tumors in a representative manner. This is a problem throughout the study. It would have been much preferable to harvest the tumors when they were around 0.5 cm³. Then, most of a section could have been quantified, avoiding the very considerable risk for bias in selecting which regions to evaluate.

6. The authors use a spontaneously transformed endothelial cell line for their biochemical analyses. It seems as if the 1G11 line lacks VEC expression however (see Fig. 4a) and it is therefore not useful for this study. The HDMEC data in Fig. S4 gives the impression that SOD3 is not expressed endogenously and only when overexpressed dramatically is there an effect on VE-cadherin levels.

7. VEC is an abundantly expressed protein that has been assigned a very important role in vessel integrity. New data from the Vestweber group has however shown that vessel integrity is maintained also in the absence of VEC expression by the endothelium (see Frye et al JEM 2015). The authors do not convince that VEC overexpression (as seen when SOD3 is induced above levels seen in the wt condition), results in increased barrier integrity. What happens with VEC-overexpressing junctions when the tissue or a culture is treated with VEGF or histamine?

8. The authors describe an important role for EC specific SOD3 in upregulation of NO resulting in the downstream effects on EC HIF-2 α and VEC transcription. Which enzyme generates the EC specific NO-induction? One would guess eNOS. Bill Sessa and his group have shown in several ambitious papers that blocking eNOS results in loss of histamine-induced vascular leakage. Thus, according to a very important line of literature, elevation of NO via eNOS activity results in leakage by causing relaxation of the vascular tone. In the current study, elevated NO is implicated in increased vascular integrity. The authors do not cite the NO literature e.g. from the Sessa lab. Please explain how these data can be integrated.

9. Throughout please quantify at least 3 independent repeats of all blots.

10. Throughout please normalize VEC mRNA and VEC staining area to a vascular marker.

Detailed criticisms:

11. Fig. 1: In the results text to Fig. 1, the authors state "Since LLC cells bear the SOD3 gene, this finding implies that Lov induced SOD3 production mainly by tumor stroma." I agree SOD3 production by the LLC cells does not seem to play a role as there is a difference between the wt and Sod3^{-/-} mice in how the tumors respond to Lov-treatment. However, what is the relative expression level of SOD3 by LLC and by the wt stroma? Fig. 1f, please quantify the data. One gets the impression that SOD3 is not normally expressed by endothelial cells also from the HDMEC data in Fig. S4

12. Fig. 2: Fig. 2a, normalize to CD31+ vessel area.

13. In Fig. 2a, Lov is shown to increase the % of lectin-perfused vessels in the SOD3KO. How do the authors explain this?

14. For Fig. 2 f and g the authors state that "Morphological analyses showed that higher SOD3 levels, induced by Ad-mSOD3 injection or Lov treatment, shifted tumor vessel morphology from disconnected small clumps of CD31+ structures to elongated, less

tortuous vessels (Fig. 2f-g)". These are not morphological analyses – these are very small regions of lectin-positive areas in the tumors. Nothing can be said from this about the morphology and the claim is disturbing. The authors need to use smaller tumors, image analyses programs and thick sections to measure vessel parameters. Also the EM analyses in Fig. 2h can't be used for conclusions unless quantification is done.

15. Fig. 2j, how was the quantification and normalization done for these data? What is the significance in the Lov/wt bar in relation to? The interesting comparison is with Vhcl/wt and that cannot be a significant difference?

16. Compared to Figure 1 it seems that tumor perfusion is unusually low in Cre- mice in the Fig. 3 results even though the Doxo levels appear to be significantly higher. This seems very unlikely; please explain.

17. In Fig. 3b, is SOD3 expressed also outside the vasculature? In panels k and l, please normalize the data.

18. In Fig. 4c, VEC mRNA levels need to be normalised to a vascular marker. Please clarify which SOD3 mice are being used in Fig. 4h. Are these the EC-Tg or Ad-SOD3?

19. In Fig. 8, are the non-tumor samples adenomas? This reviewer understand the challenges in obtaining "normal" samples, but adenomas are far from normal. It would have been perhaps more useful to categorize the CRCs based on their staging. Please spilt in stage 2 and stage 4 groups which are usually the most prevalent in clinical biopsy cohorts.

Reviewer #1

We thank the referee for his/her comments

1. What is the mechanism underlying LOV increase of SOD3 expression?

To approach this question, we first determined Lov-induced SOD3 upregulation in cancer cells, leukocytes and endothelial cells (EC) isolated from LLC tumors. We found that Lov significantly upregulated SOD3 only in EC and leukocytes (new Fig. 1f), but did not induce SOD3 expression in EC or leukocytes isolated from tumor-free organs in the same mice (new Suppl. Fig. 3a, b). Moreover, Lov addition to cultured EC did not upregulate SOD3 (new Suppl. Fig. 3c). We were unable to carry out these experiments with primary lymphocytes, since they are extremely susceptible to Lov-induced toxicity *in vitro*. We interpret these data as showing that Lov induces SOD3 through an indirect mechanism that is operative in tumor conditions. SOD3 expression is repressed in fibroblasts by some inflammatory mediators such as TNF- α [Marklund. J Biol Chem 267:6696 (1992)], and Lov downregulates TNF- α in tumor-infiltrating leukocytes [Mira et al. Oncotarget 4:2288 (2013)]. Using 3T3 cells as a model, we show that TNF- α stimulation reduced SOD3 mRNA levels, and that Lov co-addition reversed this downregulation in a dose-dependent manner. Other statins, such as simvastatin, also reversed TNF- α -induced SOD3 repression, but atorvastatin did not.

These experiments provide some clues as to how Lov triggers SOD3 transcription, but raise new questions as to why specific statins have distinct effects on SOD3 induction. It is difficult to explain the discrepancies among the statins tested, although a recent study showed specificity in gene expression profiles induced by different statins in a pancreatic cancer cell line [Gbelcová et al. Sci Rep 7:44219 (2017)]. The authors suggest that these differences are associated with intracellular statin concentrations. Whether the variations we observed are due to differential bioavailability or mechanistic dissimilarities between statins requires further study; a full answer to this question needs much more time than that provided for manuscript revision, as it constitutes a research project in itself. In our study, Lov was used as a tool to upregulate SOD3, and we analyzed only the Lov effects lost in SOD3^{-/-} mice. To clarify this point, we have removed any speculation on the clinical use of statins to potentiate chemotherapy. We nonetheless hope the reviewer finds these data of sufficient interest.

2. ...Where should SOD3 be expressed to impact the tumor?...Along these lines, would loss of SOD3 in tumors be sufficient to attenuate DOXO effect? Is it sufficient to reduce SOD3 expression in the stroma? Authors need to establish decisively which are the tissue types that exhibit the change in SOD3 and later HIF2a expression.

As commented above, Lov induced SOD3 only in stromal cells (new Fig. 1f). SOD3 is nonetheless a secreted enzyme and once secreted, might diffuse in the tumor parenchyma. It could also produce a generalized reduction in reactive oxygen species, which would affect the entire tumor, including its vasculature. Identification of the SOD3-producing cell types in the tumor environment might thus not be crucial to explaining its effect on the vasculature. Intratumor injection of Ad-SOD3 caused unselected SOD3 expression in tumor and stromal cells (see new Fig. 1g), which affected the vasculature and tumor response to Doxo. The influence (or lack of same) of SOD3 compartmentalization on these effects is discussed in the revised manuscript.

3. Is the effect of LOV on SOD3 dose dependent? Likewise, are the changes in HIF2a and VEC expression levels (does dependent) correspond to the level of LOV?

Various laboratories have reported dose-dependent dual statin effects as promoters (low dose) and inhibitors (high dose) of angiogenesis [Weis et al. Circulation 105:739 (2002)]. To

avoid these dual Lov effects on tumor angiogenesis, which could disguise the SOD3 pathway reported here, we addressed the reviewer's question *in vitro*. As shown above, we found dose-dependent Lov reversal of TNF- α -induced repression of SOD3, although Lov did not trigger SOD3 expression directly. We did not analyze HIF-2 α expression, as statins might affect expression of some PHD [Thirunavukkarasu et al. *Int J Cardiol* 168:2474 (2013)] and thus affect HIF levels in a SOD3-independent manner. We hope this answers the reviewer's concern.

4. *Would other statins equally affect SOD3? Would induced expression of HIF2 α , on its own, phenocopy the changes seen upon LOV treatment?*

We show that simvastatin, but not atorvastatin, reversed TNF- α -induced repression of the SOD3 gene. The point has been discussed at the beginning of this reply.

With regarding to the query "...induced expression of HIF2 α , on its own, phenocopy the changes seen upon Lov treatment...", we show that HIF-2 α expression in EC induces VEC expression in a dose-dependent manner; this is not observed after HIF-1 α overexpression (new Suppl. Fig. 8). We also show that HIF-2 α triggered the VEC promoter in a dose-dependent manner in 3T3 cells (new Suppl. Fig. 9c). HIF-2 α expression thus phenocopies the Lov effect on VEC expression. Analysis of *in vivo* changes in the tumor vasculature after HIF-2 α induction would involve generation of an inducible, endothelial-specific HIF-2 α transgenic mouse (similar to that we produced for SOD3), an undertaking that would take more than two years. In addition, HIF-2 α overexpression in the vasculature might induce strong phenotypes, which could be independent of SOD3 (the objective of this study). For example, HIF-2 α depletion in the tumor vasculature caused a substantial, SOD3-independent reduction in VEC expression (new Suppl. Fig. 10b).

5. *What is the role of HIF1 α in this process and how does HIF2 α being selectively activated? SOD3 was previously reported to induce HIF1 α , was this possibility ruled out?*

The VEC promoter has several HRE sites and both HIF-1 α and HIF-2 α could trigger VEC transcription. Previous work nonetheless showed that HIF-2 α , but not HIF-1 α , induces VEC gene transcription in endothelial cells in normoxic and hypoxic culture conditions [Le Bras *et al.* *Oncogene* 26:7480 (2007)]. For this reason, we focused on HIF-2 α , and did not discuss this point in the previous version.

We now reproduce Le Bras *et al.*'s results in our 1G11 system and show that HIF-2 α , but not HIF-1 α , transactivates the VEC promoter in a dose-dependent manner (new Suppl. Fig. 8a, b). We show that whereas shRNA-mediated HIF-2 α silencing reduces VEC mRNA levels in 1G11 cells (new Fig. 5f), this is not observed in HIF-1 α -silenced cells (new Suppl. Fig. 8d). The basis of specific HIF-2 α activity on the VEC promoter is not known (at least to us), and might be related to cooperation with other transcription partners [Pawlus *et al.* *PLoS One* 21:8 (2013)]. We feel that resolution of this question lies outside the bounds of this study, but obviously warrants future research.

We compared 1G11-mock and 1G11-SOD3 cells and did not detect significant SOD3 induction of HIF-1 α , nor did we find increased HIF-1 α stability associated to SOD3 overexpression in normoxia (new Suppl. Fig. 9f, g); both are detected for HIF-2 α (new Fig. 6i, j).

6. *Most if not all studies were carried out in the 1G11 cell line, which does not appear to express endogenous SOD3. What is the level of SOD3 in other cell lines (i.e. HMEC-1)? Both overexpression and inactivation of SOD3 in different cell lines is required.*

We consider that, in general, endogenous SOD3 levels in a specific cell line would not limit the conclusions from experiments involving SOD3 overexpression, particularly if levels

are low in that line. In the case of 1G11 cells, we showed that they express and secrete SOD3 to the extracellular space (new Fig. 4g, lane 1; new Suppl Fig 5a, b). We now performed SOD3 silencing experiments in which negative regulation of SOD3 levels in 1G11 cells reduced VEC expression (new Suppl. Fig 6). These experiments constitute indirect evidence that 1G11 cells express SOD3.

Regarding the other cell line used, we show that SOD3 overexpression in HDMEC (human dermal microvascular endothelial cells) also upregulated VEC (new Suppl Fig 7c, d) and reduced FITC-dextran permeability of HDMEC monolayers (Suppl. Fig. 7e). HDMEC express low levels of endogenous SOD3 (Suppl. Fig 7d, bottom), particularly compared with those expressing ectopic SOD3.

Additional points:

7. Fig 1b: 3-NT panel staining in SOD3^{-/-} after LOV treatment is high, inconsistent with data shown in panel 1a.

Data in panel 1a have been replaced by those obtained with cells isolated from tumors (new Fig 1f). In any case, Lov treatment increased SOD3 mRNA levels only in WT mice. We do not see the inconsistency indicated by the referee, since 3-NT staining would be similar in tumors grown in Vhcl- and Lov-treated SOD3^{-/-} mice. As the image showed slightly less intense 3-NT staining for Vhcl-treated SOD3^{-/-} than the Lov counterpart, we have replaced this Vhcl-treated panel with another more representative image (now in Fig. 1e).

8. Fig 2a: decrease in FITC-Lectin in SOD3^{-/-} is also seen in control. Please explain

Old Fig. 2a is now Fig. 2h. The comparison of lectin-perfused vessels between Vhcl- and Lov-treated SOD3^{-/-} mice has a p = 0.3 using a two-tailed Student's t-test. Although the differences were not statistically significant, the percentage of perfused vessels tends to be higher in Lov-treated mice. As Lov might also exert SOD3-independent activities on the vasculature, we used two additional models (adenoviral expression and inducible EC-specific overexpression) to analyze direct SOD3 effects on the tumor vasculature. There is, in general, a good agreement among results in all models, although the pleiotropic actions of statins might be responsible for small changes observed in some vascular parameters studied.

9. Fig 2g and elsewhere: data shown need to be quantified and statistically calculated

Fig 2k: What is the level of VEC mRNA in SOD3^{-/-} mice?

Figure 2 has been remodeled considerably, resulting in changes in the panels. We followed the reviewer's recommendation and quantified the length, diameter and number of branches of CD31⁺ blood vessels in all tumor models analyzed, and performed statistical analyses. We consider that these data address the referee's question.

We show VEC mRNA levels in Vhcl- and Lov-treated tumors from SOD3^{-/-} mice in Fig. 2p (now normalized to CD31 mRNA levels). In addition, we show VEC mRNA levels in EC from SOD3^{-/-} and WT mice in physiological conditions; Suppl Fig. 1 illustrates that VEC mRNA levels and other endothelial markers are comparable between WT and SOD3^{-/-} mice in homeostatic conditions.

10. Fig 3c: Although tumors are smaller in transgenic mice expressing SOD3 degree of inhibition varies between the experiments. For example, the tumor size in SOD3EC-Tg shown here is equal to the size shown in the treated groups shown in 1d groups.

All tumor experiments were performed using littermates and with appropriate genotypic controls and vehicles. Immunocompetent mice were used for all tumor models, and differences in tumor size between mice on different backgrounds or receiving different treatments could thus be ascribed to treatment effects or genetic characteristics that modify

immune cell activation, leukocyte infiltration or other parameters associated with antigen recognition or immune cell activity. For instance, Lov treatment induces macrophage polarization to an anti-tumor phenotype [Mira et al. *Oncotarget*. 4:2288 (2013)], which could affect the growth of tumors implanted in Lov-treated mice, independently of SOD3.

11. Fig 4g: statistical analysis need to be included.

Old Fig. 4g is now Fig. 4l. We now show MFI quantification from three experiments, with appropriate statistical analyses (new Fig. 4m).

12. Fig 4k: the changes, seen only in one concentration, is hard to understand

Fig. 4k is now Fig. 4r. The changes observed after NO donor treatment in fact did not occur at only a single concentration. The graph shows a biphasic effect of the NO donor on VEC promoter activity; at low NO donor concentrations (1 and 5 μ M) there was a significant increase in VEC promoter activity, whereas at very high NO donor concentrations (0.5 mM) VEC promoter transactivation decreased significantly. To confirm these data, we performed new permeability experiments with 1G11 cells treated at low or high NO donor doses. In accordance with VEC promoter activity data, we observed that low NO donor dose reduced and high dose increased 1G11 monolayer permeability (Fig. 4s). In the revised manuscript, we discuss the antithetical role of NO in endothelial permeability at greater length.

13. Fig 5c: the differences shown between ChIP1 and ChIP2 per Hif2 α expression are not clear. What is the evidence for specific HIF2 binding?

As HIF-1 α does not transactivate the VEC promoter (new Suppl. Fig. 8), it was not tested in ChIP assays.

Is the binding significantly greater?

We apologize, as the Y-axis in Fig. 5c was wrongly labeled as “% of input”, and should read “signal relative to input”. The increase in HIF-2 α binding in ChIP2 using SOD3-overexpressing cells might appear low compared to levels detected in ChIP1, but in five independent experiments, we detected no HIF-2 α binding in ChIP2 in mock cells. SOD3 increase thus induces HIF-2 α binding to a VEC promoter region to which it is not bound in mock cells. HIF-2 α binding to this proximal promoter region could be thus relevant to the SOD3-induced increase of VEC transcription. Statistical analyses for all ChIP assays are now included.

What is the level of HIF1 α , which is subject to PHD regulation?

All experiments were performed in normoxia, in which HIF-1 α expression is extremely low (see below).

Would PHD inhibitors, available commercially, phenocopy the effect of LOV?

We do not understand this reviewer’s concern here, as Lov was not used in these experiments.

14. Fig 6h: degree of ubiquitination claimed for is not compelling. Data per the half-life of HIF2 α as well as HIF1 α is needed.

We analyzed HIF-1 α and HIF-2 α half-lives experimentally after exposure of hypoxia-cultured 1G11-mock and -SOD3 cells to normoxic conditions; these data are included in new Fig. 6i, j (HIF-2 α) and new Suppl Fig. 9f, g (HIF-1 α). In hypoxia, 1G11-mock and -SOD3 showed comparable HIF-1 α and HIF-2 α levels. HIF-1 α is degraded rapidly in normoxia in both 1G11-mock and -SOD3, and no differences were detected. The HIF-2 α half-life was

slightly (but significantly) extended in 1G11-SOD3 vs. -mock cells after re-oxygenation. This distinct behavior of HIF-1 α and HIF-2 α in 1G11-SOD3 cells might be associated with the relative HIF-2 α resistance to degradation at oxygen levels that usually cause HIF-1 α proteolysis [Lofstedt et al. Cell Cycle 6:919 (2007)]. HIF-2 α is also hydroxylated much less efficiently than HIF-1 α , which might also cause preferential HIF-2 α stabilization at higher oxygen tensions [Koivunen et al. J Biol Chem 279:9899 (2004)].

15. Fig. 6i: assessment of HIF1 α is needed

Fig. 6i (now Fig. 6k) shows HIF-2 α staining in Vhcl- and L-NMMA-treated 1G11-SOD3 cells cultured in normoxia. HIF-1 α levels in 1G11 cells cultured in normoxia is very low (see representative images of HIF-1 α staining in 1G11-Mock and -SOD3 cells; for reviewer perusal), and we thus do not expect to see any L-NMMA treatment effect. This HIF-1 α staining is cited in the text as data not shown.

16. Fig 7 c-h: what are the changes upon inducible HIF2 α vs. HIF1 α expression? Inclusion of the PHD inhibitor?

We understand the interest of this point, but clarify that the data in Fig. 7 are derived from inducible, EC-specific HIF-2 α deletion (not expression). EC-specific HIF-1 α -deficient mice reportedly show reduced tumor growth and vascularization due to attenuated VEGF and VEGFR2 expression in EC [Tang et al. Cancer Cell 6:485 (2004)]. The authors also show that HIF-2 α levels are not regulated to compensate HIF-1 α deficiency, which suggests divergent HIF-2 α and HIF-1 α function in EC biology *in vivo*. Indeed, HIF-2 α is unable to compensate the severe vascular defects in embryos that lack HIF-1 α [Carmeliet et al. Nature

394:485 (1998); Iyer et al. *Genes Dev* 12:149 (1998)]. Given these reports of divergent *in vivo* HIF-1 α and HIF-2 α functions, and that our data do not support a SOD3 role in HIF-1 α stability or function in EC, we consider that EC-specific, inducible HIF-1 α deletion is unrelated to the main topic of our study (to understand how SOD3 affects tumor-associated blood vessels). In addition, we have no HIF-1 α floxed mice available.

The reviewer also suggests use of a PHD inhibitor as an alternative tool to stabilize HIF proteins, the opposite of the strategy we used in Fig. 7. We envision some problems in the implementation and interpretation of such an experiment. First, as far as we know, there are no public, solid pharmacodynamics data for the IOX2 inhibitor in mice; without this information it is difficult to determine the optimal IOX2 dose and administration route that reliably inhibits PHD. A second consideration is that systemic administration of a PHD inhibitor could have uncontrolled, whole-body effects due to non-specific, chronic HIF stabilization in various tissues. This would be especially relevant in the immune system, since the HIF-1 α /HIF-2 α balance controls T lymphocyte and myeloid cell differentiation [Palazon et al. *Immunity* 41:518 (2014)]. Given the role of myeloid and Treg cells on neoangiogenesis and immune response, it is difficult to predict the effect of IOX2 administration in tumor progression. Finally, and more important, HIF-2 α has inverse functions in EC and tumor cells, boosting tumor growth and angiogenesis when stabilized in tumor cells [Cho et al. *Nature* 539:107 (2016); Chen et al. *Nature* 539:107 (2016)], but triggering tighter EC barrier (including VEC upregulation) and pericyte association when stabilized in EC [as reported in EC-specific PHD2^{+/-} mice; Mazzone et al. *Cancer Cell* 136:839 (2009); Leite de Oliveira et al. *Cancer Cell* 22:263 (2012)]. Indiscriminate inhibition of PHD enzymes in tumor and endothelial cells might thus generate opposite effects, making interpretation of the results difficult.

17. Fig 8g,j: quantification with respective statistical analysis is required.

Here we do not understand the reviewer's request. Figure 8g shows representative images of the TMA analyzed with anti-SOD3 and -HIF-2 α antibodies. All images in the TMA were quantified, and separate H-scores (indicated in Methods) were calculated for each protein and tumor compartment (tumor cells and stroma); H-score data are shown in Fig. 8h and 8i, with appropriate statistical analyses (determination of the correlation coefficient).

Figure 8j shows representative immunofluorescence images of tumor sections stained with anti-SOD3, -HIF-2 α and -CD34 (endothelial marker). The images were quantified as a percentage of SOD3-stained CD34⁺ cells and as the percentage of SOD3-stained cells with nuclear HIF-2 α accumulation; these data were indicated in the text (we considered that a graph would add no value to the data). The data in Fig. 8j describe an observation in tumor samples, and we do not understand how to apply a statistical method as different situations are not compared.

Minor points:

More detailed legends are needed throughout the ms; often it is hard to understand what is being shown.

Although we have to adhere to the journal's 350-word limit for each figure legend, we tried to include more details and hope this better describes the experiments.

Reviewer #2

We thank the referee for his/her comments

1. One main question is whether SOD3 regulates tumor endothelium alone or also normal endothelium.... As SOD3 is secreted it would also be surprising if the effects were so compartmentalized. It would have been preferable ... if authors had started their study with characterizing the phenotype of the Sod3^{-/-} mouse.

We assumed that since SOD3^{-/-} mice do not display any overt phenotype in baseline conditions, their vasculature would show no defects compared to that of WT mice, but this was not addressed experimentally. We now show that expression of 13 endothelial markers was unchanged in WT and SOD3^{-/-} mice, as determined by quantitative PCR (new Suppl. Fig. 1).

2. In the absence of the Lovastatin treatment, there is not difference in the expression level of SOD3 in wt and SOD3^{-/-} mice (Fig. 1a). This is confusing.... The authors could do a better job here in distinguishing different pools of SOD3 in the tumors... One gets doubtful whether SOD3 is expressed in endothelial cells at all?

We measured SOD3 mRNA levels in the main cell types that compose tumor tissue in Vhl- or Lov-treated WT and SOD3^{-/-} mice. We used cell sorting to purify tumor cells (LLC-GFP cells in these experiments), endothelial cells (EC; CD31⁺) and hematopoietic cells (CD45⁺), and found that Lov induced SOD3 expression in EC and leukocytes, but not in tumor cells (new Fig. 1f). Lov did not induce SOD3 expression in EC or CD45⁺ cells isolated from tumor-free organs from the same mice (new Suppl. Fig 3), which suggests that Lov induces SOD3 in a tumor-dependent manner. Very preliminary data suggest that Lov attenuates the activity/expression of some inflammatory mediators that repress SOD3 expression.

As the reviewer indicates in his/her previous comment, SOD3 is a secreted enzyme that could diffuse (at least at short range) in tumor tissue. Identification of the SOD3-producing cell types in the tumor tissue might thus be less critical in explaining its effect on the vasculature. Intratumor injection of Ad-SOD3, which infects and triggers SOD3 expression in tumor and stromal cells (see new Fig. 1g), was sufficient to increase Doxo levels (Fig. 1i), enhance blood vessel perfusion (now Fig. 2a), and promote changes in VEC expression (now Suppl. Fig. 4c, d).

The importance (or lack) of SOD3 compartmentalization on its effects on the tumor vasculature is now discussed in the revised manuscript.

3. The LLC tumors seem to respond better to Doxo in the presence of SOD3 but overall, SOD3-positive are dependent on Lovastatin cotreatment. Lova upregulates SOD3 but it must have a variety of other effects. This should be discussed.

We concur that Lov effects are pleiotropic, and for that reason used two genetic approaches to upregulate SOD3 in tumors (intratumor injection of recombinant adenovirus and inducible SOD3 expression in EC); in these models we did not use Lov co-treatment. A paragraph has been added (Discussion) to indicate that Lov has pleiotropic effects. Length restrictions do not permit an expanded discussion of the pleiotropic effects of statins.

4. The authors claim that SOD3 mediates increased VEC and stabilized junctions. Nowhere do the authors show the morphology of junctions with sufficiently high resolution whether in the absence or presence of SOD3, in normal vessels or in the tumor vasculature. Please show

and quantify upregulation of VEC immunostaining in vivo in WT and SOD3^{-/-} mice and when SOD3 is overexpressed in ECs.

Images were acquired with a 63x objective, the magnification used for such images in other studies, as it offers a broad view of VEC staining in the tumor. We are aware that, at this magnification, junction morphology could be compromised, but preferred to reinforce the idea (also seen by mRNA quantification) that SOD3 upregulated VEC, rather than showing detailed morphological alterations of the junctions. In the revised version, we used image software to quantify the length of VEC-stained regions in our images, as described in several studies of other EC junctional proteins [e.g., see Mazzone et al. *Cancer Cell* 136:839 (2009)]. The values are indicated in the text and show that VEC⁺ junctions in tumor vessels extended over longer distances when SOD3 was upregulated. We now provide larger images and included a new panel showing only VEC staining in a magnified region of the merge image shown for all tumor models analyzed; we think that this gives a more precise idea of changes in VEC expression. We revised the text to avoid reference to morphological stabilization of junctions and focus only on the conclusions supported by our data, that SOD3 induced VEC upregulation and reduced vessel permeability *in vitro* and *in vivo*. Nevertheless, if VEC is upregulated it seems to us logical to speculate on the possibility that EC barrier is tighter.

Since VEC expression does not change between WT and SOD3^{-/-} mice in basal conditions (Suppl. Fig. 1), we understand that the referee requests VEC immunostaining in tumor vessels. We provide now images for VEC staining in tumors from Vhcl-treated WT and SOD3^{-/-} mice (new Fig. 2q; quantification in Fig. 2r). VEC staining in tumor vessels from SOD3^{EC-Tg} mice is now shown in Fig. 3m (quantification in Fig. 3n). We provide new images, at higher magnification, of 1G11-mock and -SOD3 cells stained with an anti-VEC antibody (Fig. 4c), and quantified junctional length (Fig. 4d) and the gap index in these cells (Fig. 4e).

5. The LLC tumors are grown too large. In some cases, they reach 5 cm³, which is not acceptable ethically, moreover, how do the authors quantify such large tumors in a representative manner... It would have been much preferable to harvest the tumors when they were around 0.5 cm³. Then, most of a section could have been quantified, avoiding the very considerable risk for bias in selecting which regions to evaluate.

We apologize, as this point was not well explained in the previous version. With the exception of CD31 staining (performed with end-point tumors), analyses involving VEC staining, tumor perfusion, and vessel permeability were performed with tumors excised at days 16-20 after tumor cell inoculation; at this time, tumors were usually smaller than 1 cm³. We now indicate the time at which tumors were obtained for each analysis and section thickness (in figure legends and in Methods). To avoid bias in the evaluation, we imaged various slides from the same tumor and at least four areas/slide; in many cases, an independent researcher performed blind microscopic evaluation. The reviewer suggests use of 0.5 cm³ tumors, but vascular abnormalization is typical of advanced tumors; use of very small tumors might not yet manifest abnormalization of the tumor vasculature.

6. The authors use a spontaneously transformed endothelial cell line for their biochemical analyses. It seems as if the 1G11 line lacks VEC expression however (See Fig. 4a) and it is therefore not useful for this study.

Here we cannot agree with the referee's view. We showed that 1G11 cells express VEC by immunostaining (now Fig. 4c), immunoblot (now Fig. 4a) and quantitative PCR (now Fig. 4b). As indicated above, we replaced the immunofluorescence images in Fig. 4a, and quantified junctional length and gap index. We hope the quality of the new images will dissipate any doubts about VEC expression by 1G11 cells.

7. *The HDMEC data in Fig. S4 gives the impression that SOD3 is not expressed endogenously and only when overexpressed dramatically is there an effect on VE-cadherin levels.*

HDMEC cells express low levels of SOD3, but they do express it. We added a new panel in now Suppl Fig. 7d that shows the immunoblot of the filter with an anti-SOD3 antibody. In any case, the low SOD3 levels in HDMEC cells would not affect our conclusions based on SOD3 overexpression in this cell line.

8. *New data from the Vestweber group has however shown that vessel integrity is maintained also in the absence of VEC expression by the endothelium.-The authors do not convince that VEC overexpression (as seen when SOD3 is induced above levels seen in the wt condition), results in increased barrier integrity. What happens with VEC-overexpressing junctions when the tissue or a culture is treated with VEGF or histamine?*

We analyzed the *in vitro* permeability of 1G11-mock and -SOD3 monolayers after VEGF exposure (50 ng/ml). The results (new Fig. 4h) show that VEGF increases 1G11-mock monolayer permeability, but not that of 1G11-SOD3 cells. We now show representative images of VEC-stained, VEGF-treated 1G11-mock and -SOD3 cells (new Fig. 4i). Quantification of these images showed that whereas VEGF decreases junctional length and increases the number of gaps in 1G11-mock cells, it does not affect these parameters in 1G11-SOD3 cells (new Fig. 4j, k). These results are consistent with the hypothesis that SOD3 increases barrier integrity through VEC induction (or at least in part through this mechanism).

9. *The authors describe an important role for EC specific SOD3 in upregulation of NO resulting in the downstream effects on EC HIF-2 α and VEC transcription. Which enzyme generates the EC specific NO-induction? One would guess eNOS. Bill Sessa and his group have shown in several ambitious papers that blocking eNOS results in loss of histamine-induced vascular leakage. Thus, according to a very important line of literature, elevation of NO via eNOS activity results in leakage by causing relaxation of the vascular tone. In the current study, elevated NO is implicated in increased vascular integrity. The authors do not cite the NO literature e.g. from the Sessa lab. Please explain how these data can be integrated.*

The dual NO role in vascular function was discussed briefly in the Introduction. Data in Fig. 4r (previously 4k) showed that at low concentrations (1 and 5 μ M), the NO donor DETA-NONOate triggered VEC promoter activity, whereas high doses (0.5 mM) repressed the promoter. We expanded these data using *in vitro* permeability assays, in which we found that low DETA-NONOate doses reduce and high doses increase 1G11 monolayer permeability (new Fig. 4s). Our results thus do not contradict those from the Sessa lab, and suggest that precise regulation of NO levels could have distinct effects on vessel permeability (at least *in vitro*). We now discuss this dual NO activity in more detail and include the reference from Sessa's lab. We are restricted by the journal's length limits, but hope the referee finds this discussion appropriate.

10. *Throughout please quantify at least 3 independent repeats of all blots.*

Three replicas of all blots have been quantified, and ratio data are now shown as mean \pm SEM of the densitometric values.

11. *Throughout please normalize VEC mRNA and VEC staining area to a vascular marker.*

The data for the VEC staining area were already normalized to the CD31 endothelial marker-stained area. VEC mRNA data have now been normalized to CD31 throughout the manuscript.

Detailed criticisms:

12. *Fig. 1: In the results text to Fig. 1, the authors state “Since LLC cells bear the SOD3 gene, this finding implies that Lov induced SOD3 production mainly by tumor stroma.” I agree SOD3 production by the LLC cells does not seem to play a role as there is a difference between the wt and Sod3^{-/-} mice in how the tumors respond to Lov-treatment. However, what is the relative expression level of SOD3 by LLC and by the wt stroma? Fig. 1f, please quantify the data. One gets the impression that SOD3 is not normally expressed by endothelial cells also from the HDMEC data in Fig. S4*

As mentioned above, we changed old Fig. 1a to accommodate the results of SOD3 analysis in purified tumor cell types (now in Fig. 1f). From these data, it is clear that EC do express SOD3; indeed, Lov induced SOD3 expression in EC cells (in addition to leukocytes). To determine the relative contribution of each cell type to the level of SOD3 in the tumor tissue is nonetheless a difficult task, since although neoplastic cells do express low SOD3 mRNA levels, they are more numerous than stromal cells. New immunofluorescence data show there is a tendency for slight SOD3 accumulation near blood vessels (new Fig. 1d). The reason for this pattern is not known, although might reflect preferential SOD3 binding to specific elements of the extracellular matrix near vessels or to the endothelial cell membrane.

Since SOD3 is a secreted enzyme, it could nonetheless act locally, but not necessarily in an autocrine manner. We do not think compartmentalization of SOD3 expression would be critical for its effects on tumor vasculature. Indeed, ubiquitous SOD3 expression after adenoviral injection in a Sod3^{-/-} background also improved Doxo delivery and vascular parameters. Based on these new data, we have rewritten the sentence indicated by the reviewer. Old Fig. 1f (now 1g) was quantified using the ImageJ tool to measure total fluorescence; we hope that this responds to the reviewer's request.

13. *Fig. 2: Fig. 2a, normalize to CD31⁺ vessel area.*

The original data were normalized to CD31⁺ vessel area; this is now indicated in the y axis.

14. *In Fig. 2a, Lov is shown to increase the % of lectin-perfused vessels in the SOD3KO. How do the authors explain this?*

Old Fig. 2a is now Fig. 2h. The increase in lectin-perfused vessels in the Lov-treated Sod3^{-/-} mice is not statistically significant vs. those of Vhcl-treated mice. There nonetheless seems to be a tendency to higher perfusion in this group. A possible explanation is that Lov administration has SOD3-independent effects on the tumor vasculature, which triggers this partial increase in perfusion.

15. *For Fig. 2 f and g the authors state that “Morphological analyses showed that higher SOD3 levels, induced by Ad-mSOD3 injection or Lov treatment, shifted tumor vessel morphology from disconnected small clumps of CD31⁺ structures to elongated, less tortuous vessels (Fig. 2f-g)”. These are not morphological analyses – these are very small regions of lectin-positive areas in the tumors. Nothing can be said from this about the morphology and the claim is disturbing. The authors need to use smaller tumors, image analyses programs and thick sections to measure vessel parameters.*

We rewrote the sentence and avoid claims for vascular morphology. Images from lectin⁺ vessels were obtained from 50 μm sections, and those from CD31⁺ vessels using 20 μm

sections. We quantified length, diameter and number of branches using CD31⁺ images in all tumor models, and data were analyzed statistically.

Also the EM analyses in Fig. 2h can't be used for conclusions unless quantification is done.

We analyzed the literature in depth and find no article that quantifies SEM images; we would appreciate the reviewer's advice on this point. We have nonetheless softened our conclusions based on these images.

16. Fig. 2j, how was the quantification and normalization done for these data? What is the significance in the Lov/wt bar in relation to? The interesting comparison is with Vhcl/wt and that cannot be a significant difference?

Old Fig. 2j is now Fig. 2o. The data show the ratio between fluorescence intensity of extravasated dextran and perfused vessel counts. In the revised figure, we include more data and make the comparisons according to the reviewer's suggestion.

17. Compared to Figure 1 it seems that tumor perfusion is unusually low in Cre⁻ mice in the Fig. 3 results even though the Doxo levels appear to be significantly higher. This seems very unlikely; please explain.

All tumor experiments were performed using littermates, with appropriate genotypic controls and vehicles. Differences among the parameters measured between mice on distinct backgrounds or different treatments should be taken with caution, since they could be due to treatment or genetic effects on immune cell activity. We nevertheless coincide with the referee that perfusion in the Cre⁻ group was very low; the reason is that some tumors in Cre⁻ mice had a very small number of perfused vessels (after evaluation of various slides/tumor and five areas/slide). Doxo quantification also showed unusually high values in some mice. Other Cre⁻ and Cre⁺ mice analyzed simultaneously using the same reagents showed perfusion and Doxo data closer to anticipated values; hence, we have no reason to think that the tumors with unusual values were outliers due to a technical problem. We have determined Doxo levels and perfusion in a small group of new mice to increase experimental replicas; these data have been added to those recorded previously (new Fig. 3d and f).

18. In Fig. 3b, is SOD3 expressed also outside the vasculature? In panels k and l, please normalize the data.

SOD3 expression is under the control of the inducible VEC promoter, which directs protein expression specifically in EC. Nonetheless, as SOD3 is secreted to the extracellular space, it can very likely diffuse from the EC that produce it. Indeed, in Fig. 3b there appears to be some green staining (SOD3) that does not coincide with CD31 structures.

The data in panels k (now panel n) and l (now panel o) were normalized to the CD31 area and to the number of lectin⁺ vessels, respectively.

19. In Fig. 4c, VEC mRNA levels need to be normalised to a vascular marker. Please clarify which SOD3 mice are being used in Fig. 4h. Are these the EC-Tg or Ad-SOD3?

Data in panel Fig. 4c (now 4b) have been normalized to CD31 expression. We indicate that mice used in Fig. 4h (now 4n) are Lov-treated WT and SOD3^{-/-} mice. We added a new panel (Fig. 4o) to show that Lov treatment does not alter eNOS expression in these mice (in addition, Suppl. Fig. 1a shows that eNOS levels are comparable in WT and SOD3^{-/-} mice in basal conditions); differences in NO detected with the DAR-1 probe are thus not the result of differential NO production between the two groups, but of conserved NO availability.

20. In Fig. 8, are the non-tumor samples adenomas? This reviewer understand the challenges in obtaining “normal” samples, but adenomas are far from normal. It would have been perhaps more useful to categorize the CRCs based on their staging. Please split in stage 2 and stage 4 groups which are usually the most prevalent in clinical biopsy cohorts.

The non-tumor samples in Fig. 8 are not adenomas, but rather colon samples from the same patients more than 10 cm from the surgically removed primary tumor. These “normal” colon tissues were analyzed histologically by hematoxylin/eosin staining to verify that they bore no tumor cells. Those with minimal histological alteration in colon mucosa were discarded, which is the reason for the smaller number of controls compared to tumor samples. This is now stated in Methods and in the text.

Regarding the possibility of splitting patients into stage 2 and stage 4 groups, the cohort was composed of only 22 stage 2 and 17 stage 4 patients. To use only these samples would reduce the power of the statistical analysis.

Reviewers' comments:

Reviewer #1 (Remarks to the Author):

Authors have addressed most of the reviewer queries and the manuscript is clearly improved. I still have a couple of issues that require further experimental data.

1. concern that the effect studied is largely limited to a single cell line (1G11). limited data - in response to reviewer question - has been added for the HDMEC cells. Yet, it will be important to establish that the phenomenon studied in 1G11 cells is also seen in other cell lines.

2. the possible consideration for short term administration of PHD inhibitor in order to assess immediate response - per SOD3 expression along the model offered by the authors should be made. As example - please see PMID: 28805660.

Reviewer #1

We thank the referee for his/her comments

1. *Concern that the effect studied is largely limited to a single cell line (1G11). limited data - in response to reviewer question - has been added for the HDMEC cells. Yet, it will be important to establish that the phenomenon studied in 1G11 cells is also seen in other cell lines.*

In the revised manuscript, we have added new data showing increased VE-cadherin levels in SOD3-expressing primary bovine aortic endothelial cells (BAEC) compared to mock-transduced cells as determined by RT-qPCR and immunofluorescence (Suppl. Fig. 7g, h). We also show that SOD3 expression reduced dextran permeability of BAEC monolayers in a NO-dependent manner (Suppl. Fig. 7h). These results are similar to those obtained in 1G11 and HDMEC cells.

Another important phenomenon reported in this study is the SOD3 effect on HIF-2 α stability. We show in new Suppl. Fig. 9c-e that ectopic SOD3 expression in HDMEC cells increased the percentage of nuclear area as well as the fluorescence intensity of HIF-2 α staining compared to mock-transduced cells. We attempted similar analyses in BAEC using two polyclonal anti-HIF-2 α antibodies (NB100-122 NovusBiologicals, the same used for HDMEC, and 119-14272 RayBiotech); neither antibody recognized bovine HIF-2 α in immunoblot (not shown). We also tested the RayBiotech antibody for HIF-2 α staining in BAEC, as the company claims it recognizes bovine HIF-2 α by immunofluorescence. The staining pattern (Fig. A, below) was irregular and inconsistent; in many cases only a fraction of the cytoplasm was stained. This odd staining pattern and the lack of HIF-2 α detection in immunoblot raises –in our view– serious doubts as to the specificity of this antibody; for that reason, we prefer not to use these images in our study (although image quantification showed significant enrichment of the stained nuclear area in SOD3-expressing BAEC). We hope the reviewer agrees with this decision.

2. *The possible consideration for short term administration of PHD inhibitor in order to assess immediate response - per SOD3 expression along the model offered by the authors should be made. As example - please see PMID: 28805660.*

We performed the experiment suggested by the reviewer according to the reference provided. Tumor-bearing C56BL/6 mice were treated with molidustat (5 mg/Kg i.p.) or vehicle at day 12 post-tumor cell inoculation (mean tumor size 0.7 cm³), and 2 h later with doxorubicin (2.5 mg/Kg i.p.); mice were euthanized 20 h later and the tumor excised for analysis of the molidustat effect on HIF-1 α /2 α and doxorubicin levels. With this schedule, we found that molidustat treatment did not enhance HIF-2 α protein levels compared to controls, as determined by immunoblot (Fig. Ba for reviewer). We nonetheless observed a slight but consistent increase in HIF-1 α protein levels in molidustat-treated tumors (Fig. Bb, c).

Immunofluorescence analysis of HIF-1 α /2 α in tumor sections was consistent with immunoblots; i.e., there were no clear changes in HIF-2 α but staining was slightly stronger for HIF-1 α in tumors from molidustat-treated mice compared to controls (Fig. C). This increase of HIF-1 α suggests that molidustat had some inhibitory effect on the PHD activity in the tumors. Finally, doxorubicin levels were significantly higher in vehicle- than in molidustat-treated tumors (Fig. D). As commented in our previous reply, indiscriminate inhibition of PHD enzymes in tumor, endothelial, and immune cells might generate opposite effects, making interpretation of the results difficult. Moreover, HIF-1 α and HIF-2 α have opposite effects on tumor vasculature remodeling. For instance, the increase in HIF-1 α levels associated to molidustat treatment might cause direct or indirect induction of angiogenic and other factors involved in vascular abnormalization. Independently of the means by which molidustat reduced doxorubicin levels (outside the scope of the present study), we consider that the lack of effect of molidustat treatment on HIF-2 α in our experimental setting does not permit conclusions on the HIF-2 α effect on doxorubicin delivery into tumors. We thus prefer not to include these data in the manuscript, as they could confuse the reader rather than clarifying the role of SOD3 in chemotherapeutic drug delivery. We hope the reviewer shares this view.

Fig. B for reviewer. *a, b*) Representative immunoblots of lysates from tumors developed in vehicle- or molidustat-treated mice, hybridized with anti-HIF-2 α (*a*) and -HIF-1 α (*b*) antibodies; tubulin was used as loading control. *c*) Densitometric analysis of immunoblots as above; the HIF-1 α or -2 α /tubulin ratio was calculated. Data shown as mean \pm SEM ($n = 5$ /group).

Fig. C for reviewer. Representative histological sections from tumors developed in vehicle- or molidustat-treated mice, stained for CD31 (red), HIF-2 α (green) and HIF-1 α (far red; pseudocolored blue); nuclei were counterstained with DAPI (blue).

Fig. D for reviewer. Quantification of doxorubicin levels in tumors from vehicle- and molidustat-treated mice. Data shown as mean \pm SEM ($n = 6$). **, $p < 0.01$, two-tailed Student's t -test.

Note: doxorubicin levels are lower in this experiment than those reported in other experiments along the manuscript, since these mice received only a single doxorubicin injection.

REVIEWERS' COMMENTS:

Reviewer #1 (Remarks to the Author):

Authors have addressed key concerns that were raised by the reviewer, and at this point, I think the manuscript has been strengthened and makes a more compelling case for the role of SOD3 in chemotherapy response, and the role of HIF2a in this pathway.

Reviewer #1

- 1. Authors have addressed key concerns that were raised by the reviewer, and at this point, I think the manuscript has been strengthened and makes a more compelling case for the role of SOD3 in chemotherapy response, and the role of HIF2a in this pathway.*

We thank the referee for the positive evaluation of our work and for his/her helpful comments throughout the review process.